# Connection between the length of day and wind measurements in the mesosphere and lower thermosphere at mid and high latitudes.

Sven Wilhelm[1], Gunter Stober[1], Vivien Matthias[2], Christoph Jacobi[3], and Damian J. Murphy[4]

[1]Leibniz Institute of Atmospheric Physics at the University of Rostock, Kühlungsborn, Germany

[2]Potsdam Institute for Climate Impact Research, Potsdam, Germany

[3]Universität Leipzig, Institute for Meteorology, Germany

[4]Australian Antarctic Division, Kingston, Tasmania, Australia

**Correspondence:** S. Wilhelm (wilhelm@iap-kborn.de)

**Abstract.**

This work presents a connection between the density variation within the mesosphere and lower thermosphere (MLT) and changes in the intensity of solar radiation. On a seasonal time scale, these changes take place due to the revolution of the Earth around the Sun. While the Earth, during the northern hemispheric winter, is closer to the Sun, the upper mesosphere expands due to an increased radiation intensity, which results in changes in density at these heights. These density variations, i.e. a vertical redistribution of atmospheric mass, have an effect on the rotation rate of Earth's upper atmosphere owing to angular momentum conservation. In order to test this effect, we applied a theoretical model, which shows a decrease of the atmospheric rotation speed of about $\sim$4 m/s at a latitude of $45°$ in the case of a density change of 1% between 70 and 100 km. To support this statement, we compare the wind variability obtained from meteor radar (MR) and MLS satellite observations with fluctuations in the length of a day (LOD). Changes in the LOD on time scales of a year and less are primarily driven by tropospheric large scale geophysical processes and their impact on the Earth's rotation. A global increase of lower atmospheric eastward directed winds leads, due to friction with the Earth's surface, to an acceleration of the Earth's rotation by up to a few milliseconds per rotation. The LOD shows an increase during northern winter and decreases during summer, which corresponds to changes in the MLT density due to the Earth-Sun movement. Within the MLT the mean zonal wind shows similar fluctuations as the LOD, on annual scales as well longer time series, which are connected to the seasonal wind regime, as well as, to density changes excited by variations in the solar radiation. A direct correlation between the local measured winds and the LOD on shorter time scales cannot clearly be identified, due to stronger influences of other natural oscillations on the wind. Further, we show that, even after removing the seasonal and 11-year solar cycle variations, the mean zonal wind and the LOD are connected, by analyzing long-term tendencies for the years 2005 - 2016.

# 1 Introduction

According to the first Kepler's law, the Earth travels in a good approximation on an elliptic trajectory around the Sun. Within a year the distance between both celestial bodies changes. During the northern hemispheric (NH) winter the range is approximately 3.29% shorter than in the NH summer. Due to the inverse square law, where the intensity $I$ of the radiation is inversely proportional to the Earth-Sun distance squared this shorter distance between the Sun and the Earth during boreal winter leads to an increased heating of the mesosphere and lower thermosphere (MLT) resulting in an expansion of the MLT and thermosphere, compared to the annual mean. Another effect on the expansion/shrinking of the MLT is given by the variability of solar radiation due to the 11-year solar cycle effect. Figure 1 shows a scheme of the Earth-Sun constellation and the resulting effects, which will be explained in the following. Previous studies as, e.g., Walterscheid (1989), Marsh et al. (2007), Emmert (2015), and Lee et al. (2018) showed that solar cycle variations affect the atmospheric density, temperature, chemical composition and winds over the whole atmosphere, but in particular, in the MTI (Mesosphere-Thermosphere-Ionosphere) system. A model simulation by (Marsh et al., 2007) showed, for the whole atmosphere, response to changes in the 11-year solar cycle, with e.g., the result of temperature changes in the lower thermosphere by over 100 K at solar maximum relative to solar minimum. Further, they showed the occurrence of tropospheric wind and temperature changes due to changes in solar radiation. But they also mention that changes in the climatology due to solar radiation are too complex to be explained by simplified methods. Stober et al. (2014) showed that a solar cycle effect between 2002 and 2013 led to changes in the neutral density within the MLT region by up to 2.5%. Furthermore, satellite measurements showed on global scales a neutral density decrease by up to ∼30% between solar maximum and solar minimum at about 400 km (Emmert et al., 2010). For the winter season 2009/2010 Stober et al. (2012) showed a connection between the neutral density and the expansion/shrinking of the atmosphere by using meteor radar (MR) winds, Lidar, and Microwave Limb Sounder (MLS) satellite temperature measurements. Further, they showed a strong anti-correlation of neutral air density and prevailing zonal winds. This indicates that an increase/decrease of the neutral density occurs almost simultaneously with a decrease/increase in zonal wind speed, respectively.

Changes in the thickness of the atmosphere, resulting from differences in the distance between Earth and Sun as well as from solar cycle effects, go along with changes of the Earth's rotation speed. Based on the conservation of angular momentum $L$, the angular velocity $\omega$ for an altitude defined atmospheric layer $a$, with the thickness $a_o - a_i$, can be estimated by:

$$L = J\omega = \frac{2}{5}\, m\, \frac{a_o{}^5 - a_i{}^5}{a_o{}^3 - a_i{}^3}\, \omega, \tag{1}$$

where $J$ is the moment of inertia for a spherical shell, which rotates about an axis through the center, $a_{o,i}$ are the inner and outer radius of the spherical shell, and $m$ is the atmospheric mass. On this occasion the atmospheric mass is calculated according to Trenberth and Guillemot (1994) by

$$m = \int\limits_{r_0}^{\infty} \int\limits_{0}^{2\pi} \int\limits_{-\pi/2}^{\pi/2} \rho r^2 cos\phi \; d\phi \; d\lambda \; dr, \tag{2}$$

where $\rho = \rho(\lambda, \phi, r)$ is the density of air at longitude $\lambda$ and latitude $\phi$, and $r$ is the distance from the Earth's center, while $r = r_0$ at the surface of the Earth. In a good approximation the Earth's surface can be described as an ellipsoid $r_0^2 = a^2(1 - 2\alpha sin^2\phi)$, where $a$ is the equatorial radius, $\alpha = (a^2 - b^2)/2a^2$ is related to the flattening and $b$ is the polar radius. With respect to the height above the surface $z$, this results in $r^2 = (a+z)^2(1 - 2\alpha sin^2\phi)$ and $dr = (1 - 2\alpha sin^2\phi)^{\frac{1}{2}}dz$. Further, under the assumption that $\rho = \rho_1(r)\rho_2(\lambda, \phi)$, the atmospheric mass can be derived by

$$m = \int\limits_{0}^{2\pi} \int\limits_{-\pi/2}^{\pi/2} \left[ \int\limits_{0}^{\infty} \rho_1(z)(a+z)^2 dz \right] \rho_2(\lambda, \phi)(1 - 2\alpha sin^2\phi)^{\frac{3}{2}} cos\phi \; d\phi \; d\lambda \quad . \tag{3}$$

The integral with respect to $z$ and the relation to the measurements of the surface pressure $p_s$ can be estimated by solving

$$p_s = \int\limits_{0}^{\infty} \rho_1(z)g(z) \; dr, \tag{4}$$

where $g$ is the acceleration due to gravity. Considering that $g$ is a function of height and latitude the total atmospheric mass can be written in numerical terms as $m = 5.22371 \text{ x } 10^{15} \; \bar{p}_s$, where m is given in kilograms, and $\bar{p}_s$ is given in hectoPascal,

for standard gravity at $45°$ latitude. More detailed information about the estimation of the total mass of the atmosphere can be found in Trenberth and Guillemot (1994). According to Trenberth and Smith (2004) the total mean mass of the atmosphere is $5.148 \times 10^{18}$ kg and varies slightly on annual scales mainly due to the amount of available water vapor.

A method to measure variations in the rotation speed of the solid Earth is estimating the time the Earth needs for a full rotation. In the following, we define the crust, mantle, and core of the Earth as solid Earth. To estimate the percentage of the atmospheric

rotation velocity from the solid Earth rotation velocity, their rotation rate, and their variations are necessary. The time the Earth needs for a full rotation is not constant. The rate of rotation and the orientation of the Earth's axis varies in time and space. Perturbations in the Earth's rotation rate are caused either by external forces, as e.g., the influence of celestial bodies, or by internal torques, which are, e.g., large scale geophysical processes (Brzezinski et al., 2001). These internal torques are a combination of relative movements and mass reallocation of Earth's core, mantle, crust, oceans tides, and the atmosphere.

Geographical differences in wind pattern and oceans cause shifts in the air and in the water masses. Earthquakes displacing the Earth's mantle might also influence the Earth's rotation on longer time scales (Carter and Robertson, 1986).

On time scales less than a year the dominant geophysical process to influence the duration of the Earth's rotation is the atmosphere (Volland, 1988). Every large scale momentum exchange of the Earth's atmosphere on the Earth's surface increases or decreases their rotation, due to the law of conservation of total angular momentum within its system. The total angular momentum of the Earth's atmosphere $M$ can be approximately written as

$$M = \int_v \rho_{apc} L_{apc}\, dV = \int_v \rho_{apc} r \times (u_{rel} + \omega \times r)\, dV, \tag{5}$$

where $L_{apc}$ is the angular momentum of an air parcel of unit mass, $\rho_{apc}$ the density of the air parcel, $u_{rel}$ the relative velocity, and $\omega \times r$ is the velocity due to the rotation of the Earth (Madden and Speth, 1995).

The total angular momentum and the velocities can be split into two parts. The mass part $M_\omega$ represents the value the angular momentum would take if the atmosphere were vertically and horizontally stationary relative to the ground, and the relative part $M_r$ describes the part of the atmosphere angular momentum that is due to the motion of the atmosphere relative to the Earth's rotation. Following Madden and Speth (1995), Egger et al. (2007), and Driscoll (2010) these parts of angular momentum can be written as

$$M = M_\omega + M_r = \frac{r^4 \omega}{g} \int_0^{2\pi} \int_{-\pi/2}^{\pi/2} p_{sfc} \cos^3\theta\, d\theta\, d\lambda + \frac{r^3}{g} \int_0^{1000} \int_0^{2\pi} \int_{-\pi/2}^{\pi/2} u_{rel} \cos^2\theta\, d\theta\, d\lambda\, dp. \tag{6}$$

Thus, changes in the atmospheric angular momentum depend on the sum of different torques $dM/dt = T_F + T_M + others$. Here $T_F$ is the friction torque, $T_M$ is the mountain torque, and others torques include for example, the gravity wave torque, which is small compared to the other two mentioned. The friction torque is exerted on the Earth's surface mainly due to frictional forces between the wind and the surface. If eastward directed surface winds are prevailing on a global scale, this torque leads to an increase of the rotation rate due to angular momentum transfer from the atmosphere to the Earth's surface. The mountain torque is based on the surface pressure and orography, and it is the torque which is exerted on the Earth's surface due to a difference of pressure on two sides of a mountain. Both torques vary according to their global location and reach values in the range of $10^{19}$ Nm (Egger et al., 2007; Driscoll, 2010; de Viron and Dickey, 2014). The dominant exchange of the angular momentum between atmosphere and Earth takes place in the atmospheric boundary layer, which, depending on the orography and latitude, has a typical thickness of about 1 km at mid-latitudes (Volland, 1988).

Already in the 1960s and 70s scientists showed that fluctuations in the orientation of the Earth's rotation axis, on seasonal time scales, are associated with changes in the east-west tropospheric wind on a global scale and therefore accompanied with a transfer of angular momentum between the Earth's crust and the atmosphere (Munk and MacDonald (1961), Lambeck (1978)).

Changes in the speed of the Earth's rotation axis can be seen in fluctuations in the duration around a day. These fluctuations have been measured since the 60s using the Very Long Baseline Interferometry (VLBI) technique. The fluctuation in the day length is the difference between the astronomically determined duration of a full day $2\pi/D$ and the standard 86400 SI seconds, whereby $D$ is the angular velocity (Aoki et al., 1981). Henceforth, we use the acronym LOD for the fluctuations in the length of day. The LOD can be written as

$$LOD(t) = \frac{2\pi}{D} - 86400s. \tag{7}$$

Within the estimation of the LOD the sidereal time gets converted into solar time, by taking into account the Earth's position, nutation, precession and motion with respect to the stars. Detailed information about the transformation from sidereal time into solar time can be found in (e.g., Aoki et al., 1981; Schnell, 2006).

Carter and Robertson (1986) studied the influence of geophysical processes of the atmosphere on the duration of a day. They showed that when the globally averaged mean winds from east to west increase, the rotation rate of the Earth decreases and the day gets longer. Rosen and Salstein (1991) showed that the effect of the wind on the LOD decreases with heights, by showing that winds in the atmospheric layer between 1000 and 10 hPa contributes 0.5 ms, from 10 to 1 hPa contribute 0.03 ms, and winds above 1 hPa contributes less than 4 $\mu$s to the inter-annual LOD budget. The impact of large scale geophysical processes like, e.g., El Niño (e.g., Dickey et al., 1994) and the stratospheric quasi-biennial oscillation (QBO) can also be seen in the LOD (e.g., Volland (1988), Eubanks et al. (1988)).

On short time scales a change in the Earth rotation can lead to an uneven heating of the Earth's surface, which results to temperature differences between the surface and the atmosphere above. This can further cause convection currents, which leads to pressure differences in the atmosphere and results in a different wind formation, which can influence the day length. On a longer time scale and especially on higher altitudes increases the importance of the solar influence. An increase of the solar radiation, which can be caused due to a slowing of the Earth's rotation, leads to an expansion of the higher atmosphere, which further results, due to the conversation of angular momentum, in a slower rotation of the atmosphere. What further needs to be considered is e.g., the influence of volcanic eruptions, which influence the Earth's rotation as well as the atmospheric chemistry/temperature (e.g., She et al., 2015). Changes in these parameters can further lead to changes in the neutral density.

Within this study, we focus on heights between 60 and 100 km. These heights are sensitive enough to density changes due to the changes in the intensity of solar radiation. After we describe the data we used in this study in Section 2, we show results and discuss the theoretical change of the rotation speed due to an expanding/shrinking atmosphere in Section 3. We will show that due to the expansion/shrinking effect even under the assumption of equal density distribution between the northern and southern hemispheres (SH), differences in the prevailing wind occur. Furthermore, we will show a connection between the LOD and the prevailing wind by showing correlations in the MLT region by using MR and MLS data for one polar and two

mid-latitude locations. We use the LOD data to show, how deep the influence of solar radiation penetrates into the atmosphere. The conclusions are found in Section 4.

## 2 Data

The wind data we use in this study are derived from MR and MLS satellite measurements. The MRs are located at the northern
high latitude station Andenes (32.5 MHz, 69.3°N, 16.0°E, Norway), the mid-latitude stations Juliusruh (32.5 MHz, 54.6°N, 13.4°E, Germany), and Collm (36.2 MHz, 51.3°N, 13.0°E, Germany) on the northern hemisphere and the southern high latitude station Davis (33.2 MHz, 68.6°S, 78.0°E, Antarctic). The radars cover an altitude range between 75 and 110 km and the obtained winds have an hourly temporal resolution and a vertical altitude resolution of 2 km in the applied analysis. At 90 km altitude, the observed volume of each radar has a diameter of approximately ~400 km, and the mean wind above each
station is a weighted average over this volume. In the case of the Andenes, Davis and Collm MR data are available between 2005 and 2016 and for Juliusruh since 2008. We focus on an altitude range between 78 and 100 km where we obtain continuous measurements. The statistical uncertainties of winds are obtained from a fitting procedure by taking into account the number of detected meteors per altitude and time bin, as well as a full non-linear error propagation of the radial wind errors. Therefore the resulting uncertainties for the hourly winds vary in a range between 2 and 6 m/s with larger errors at the upper/lower part
the of the meteor layer. More information about the all-sky meteor radars and the used wind estimation method can be found in Hocking et al. (2001), Holdsworth et al. (2004) and Stober et al. (2017). For this research, we focus primarily on the zonal wind component, because a connection between winds and changes in day length will be mainly seen in the main rotation direction of the Earth.

In addition to local radar observations, we use satellite data from the Microwave Limb Sounder (MLS) to extend the vertical
coverage. MLS onboard the Aura satellite (Waters et al. (2006), Livesey et al. (2015)) has a global coverage from 82°N to 82°S and an useful height range from approximately 11 to 90 km (261 to 0.001 hPa). The vertical resolution varies between ~4 km in the stratosphere and ~14 km at the mesopause (Livesey et al., 2007). The geometric heights are approximately estimated from pressure levels as described in Matthias et al. (2013): $h = -7 \cdot ln(p/1000)$, where $h$ is the altitude in km and $p$ the pressure in hPa. Furthermore, we are aware about a difference between the geometric and geopotential heights, which
increase especially above 80 km. Therefore, we focus in this work on the height range between 60 to 80 km (if not otherwise specified) to investigate a connection between the LOD and the density depending zonal wind within these heights. Daily quasi geostrophic winds for the years between 2005 and 2016 are derived from MLS geopotential height observations. For this study we use three different horizontal grids which are located around Andenes (70°N and 0-20°E) and around Juliusruh/Collm (50-60°N, 0-20°E), which are further referred to as northern high and mid latitude stations, respectively. For the SH we use a
horizontal grid around Davis (70°S, 60-80°E).

Further we use in this study combined data from the international Earth Rotation and Reference System Service (IERS). The use of interferometry between several stations, which observe radio sources, leads to fundamental geodetic information as changes in the Earth's spinning or in the Earth orientation (Rothacher (2002), Altamimi et al. (2007), Boeckmann (2010)).

Based on these information the mean rotation rate and the astronomical duration of the day were computed according to equation 6 (Aoki et al., 1981). The IERS provides uncertainties for the day length measurements which most of the time vary in a range of ∼5%. More information about the data provided by IERS and their algorithm to estimate the duration of a day can be found in Bizouard et al. (2017).

## 3  Results and Discussion

### 3.1  LOD and neutral air density at the MLT

Figure 2 shows composites of zonal winds from MR measurements at Andenes, Juliusruh, Collm, and Davis. These data are estimated by using a mean wind adaptive spectral filter (Stober et al., 2017). It uses a 1 day sliding window, which mainly removes the impact of short-term variations, as atmospheric tides and gravity waves. All three NH stations show almost similar wind patterns, with typical mesospheric eastward directed winds during the winter, with mean values of up to 10 m/s, and a wind reversal during spring. The spring wind reversal occurs earlier at mid latitudes than at polar latitudes. During the summer considerable vertical wind shear is present with westward directed winds below 90 km for Andenes, below 88 km for Juliusruh, and below 85 km for Collm. Above these heights a strong eastward jet occurs. The westward and the eastward jets reach wind values of up to 40 m/s at all three locations. These annual wind climatologies are consistent with previous studies e.g., Manson et al. (2004), Hoffmann et al. (2010), and Jacobi (2012). Compared to Andenes a nearly opposite wind pattern can be seen for Davis. A dominant eastward directed wind occurs between March and September for the complete observation range. Between September and March occurs a vertical wind shear, which reaches around October heights above 100 km. Compared to the NH stations the summer vertical wind shear remains more below 90 km.

Besides the radar data we additionally use MLS data within this study to extend the vertical coverage down to 60 km. In Figure 3 the zonal wind is shown for the high latitude location of Andenes, for middle latitudes at Collm and for the southern latitude location Davis. The altitude ranges between ∼60 and ∼90 km geopotential height. A comparison of the MLS composite winds with MR composite winds results in a qualitatively good agreement for the seasonal amplitudes and phases. Both NH locations show eastward directed winds between September and April for nearly all altitudes, with values of up to 40 m/s for the high latitude area and up to 60 m/s for the midlatitudes. During summer westward directed wind dominates below 95 km and reaches values of up to 30 m/s for the high latitudes. For the middle latitude, below 90 km, the wind reaches values of up to 50 m/s. A similar pattern of an eastward directed wind occurs in both cases during summer above 90 km geometric height. The SH location also shows similar wind pattern as the observed MR data. In the following discussion we will focus on the MLS altitude range 60-80 km and use the MR data for the altitudes between 80 and 100 km.

According to previous studies as e.g., by Emmert et al. (2004) and Stober et al. (2012), a connection exists between the thickness of an atmospheric layer and the density fluctuation within that layer. Stober et al. (2012) explained the occurrence of this connection by showing variations in the neutral density, based on MLS and MR observations, together with changes in the MLT geometric height. Furthermore they showed a strong anti-correlation between the simultaneous occurrence of the zonal wind and the density change within the mesosphere.

To underline this statement, we show in the following part the connection between the expanding MLT and the atmospheric rotation speed. Figure 4 shows, as an example, the theoretical variation in the atmospheric rotation velocity with height due to a density increase up to 1% between 70 and 100 km. The calculation is done in 2 km height layers and for the latitude of $45°$. Different latitudes lead to slightly different values of $g$, which is used in equation 4. The density increase takes place for longer time scales during a solar maximum (e.g., Emmert et al., 2010) and on annual time scales during the winter, when the Earth-Sun distance is smaller. Both cases influence the temperature within this atmospheric layer as well as their expansion compared to the annual mean. Overall the density variation during an 11-year solar cycle is stronger than the variation caused by the changes of the Earth-Sun distance. According to equations 1 - 4, we estimated for three different cases (linear (red), exponential (green) and a Gaussian shape (blue) density increase) the resulting theoretical change in the rotation speed within these heights, with the solid Earth rotation speed (black) as background flow. Based on the conserved quantity of the angular momentum within a narrow atmospheric layer (2 km vertical) this sums up, according to each case, to a decrease of the rotation speed by up to ∼2-4 m/s, with the strongest variation within the Gaussian shaped curve. These results fit to the observations by Stober et al. (2012) and show the dependence of the rotation speed within an atmospheric layer due to changes in the neutral density. However, only based on wind measurements we are not able to extract a specific wind value.

Based on ERA40 data, Trenberth and Smith (2004) showed that the global mean of the surface pressure is nearly constant, and surface pressure anomalies at the northern and the southern hemispheres are nearly identical, but the fluctuations are opposite in sign. These anomalies are mainly due to the changing amount of available water vapor in the atmosphere. Under the assumption of opposite surface pressure anomalies within both hemispheres and therefore by neglecting other factors as e.g., different gravity wave forcing between the hemispheres, we assume, on annual scales, similar pressure values within the MLT region. Therefore the prevailing wind within the MLT region should be similar in magnitude between Andenes and Davis, which are located at the same latitude in the northern and southern hemispheres. To underline the influence of the intensity of the solar radiation on the density and also on the amplitude of the zonal wind, we compare the evolution of the seasonal mean wind measurements from the NH station Andenes ($69.3°N$) and SH station Davis ($68.3°S$). Figure 5 shows, for both stations, the winter and summer mean wind for the altitudes at 88 and 96 km. The northern winter includes the mean of the months December, January and February, and the southern winter the months June, July and August. The northern winter period comes along with the perihelion, which is the point where the Earth comes nearest to the Sun. At the perihelion the intensity of the solar radiation on the upper atmosphere is higher as during the aphelion. While during the winter season the wind values are higher over Davis for both altitudes, they are higher over Andenes during the summer season, especially at 96 km, with values of up to 10 - 20 m/s. Both seasonal wind differences are consistent with the change in the average density within the upper mesosphere, resulting from the different distance between Earth and Sun and leading to the variation of the averaged zonal wind, as shown in Stober et al. (2012). We have to note that others factors exists, which are more dominant for the wind differences between both locations at theses altitudes. Other physical processes have also a strong effect on the hemispheric wind differences e.g., the topography, chemical composition of the atmosphere (Marsh et al. (2007), Lee et al. (2018)), and the occurrence and propagation of gravity waves. These waves are the main drivers of the atmospheric wind circulation and therefore also influence the local wind differences at both hemispheres. Furthermore, gravity waves lead, compared to the annual mean, to a colder

summer mesosphere and a warmer winter mesosphere (e.g., Lübken et al., 2014). These temperature differences also fit well to the atmospheric expansion/shrinking. Unfortunately, based only on wind measurements we are not able to estimate a precise value on how strong the connection is between mean zonal wind with the LOD. For a more detailed understanding of these phenomena global density observations would be required.

## 3.2 Correlation of mean winds and LOD

In the following we want to show that the LOD (fluctuations in the length of a day) correlates with the prevailing wind from the four stations. If the Earth's rotation is constant the LOD should be zero, however, small wobbles of the Earth's rotation between the days cause tiny fluctuations in the day length. These have to be compensated by a momentum transfer between the different
parts of the Earth including the atmosphere. As the atmosphere is slaved to the Earth crust, because the atmospheric momentum and mass are much smaller than that of the Earth core, the atmosphere has to respond to changes in the rotation velocity and at least the troposphere can trigger an own feedback on the LOD. So far we use the LOD explicitly as reference for the changes in the rotation speed, which can be seen in the zonal wind, as well as to verify up to which height the solar driven density effect is dominant. Therefore the next two Figures 6 and 7 show wind values for Andenes, Collm, and Davis at different altitudes and
the LOD by using the same filtering method as done for the winds. Two different altitudes in the MLT are considered from the MR winds for all locations: (1) 80 km, where within a year a change between eastward and westward directed wind occurs, and (2) 96 km, the altitude where the wind, during each hemispheric summer shows the opposite direction as at 80 km (see Figure 2). Positive wind values correspond to eastward directed winds and positive LOD values correspond to a longer duration of the day. If not explicitly mentioned, the results of the two mid-latitude stations are nearly identical. Therefore we only show the
results for the location around Collm.

At 80 km (Figure 6) the oscillation pattern of the smoothed zonal wind (blue) and the smoothed LOD (black) are similar for Andenes. According to previous studies the LOD consists of superpositions of several periods, as 0.5 years, 1 year (see also Vondrák and Burša, 1977), 2-3 years (Buffet, 1996), 5.9 years (Abraca del Rio et al., 1999) and others (e.g., Munk and MacDonald (1961), Holme and de Viron (2013)). According to Abarca-del Rio et al. (2003) an accurate estimation of the
25 impact of the solar radiation is quite complicated, due to the fact that internal oscillations in the climate system show variations with the same frequency as the 11 year solar cycle. Further, Gray et al. (2010) support this statement and mention that the problem is further complicated due to the small influence of the solar forcing on the climate. Nevertheless, Chapanov and Gambis (2008) showed that based on a decomposition of the LOD, the solar activity (10.47 years) is included. Also the zonal wind includes a superposition of several periods as the solar cycle, diurnal, and semidiurnal tides and more (e.g., Emmert et al.
(2010), Hoffmann et al. (2010)). Therefore, we additionally show with the red line a smoothed zonal wind after removing variations due to the 11-year solar cycle. The influence of the solar cycle on the daily zonal wind is relatively small, therefore the smoothness of the red line is enhanced for better visualization. Changes in the LOD are sluggish compared to variations in the wind, due to the amount of momentum which is needed to influence the Earth's rotation speed. According to e.g., Dickey et al. (1994), a direct effect between the stratospheric and tropospheric zonal wind and the day length exists, on annual time

scales due to long term geophysical effects, as e.g., QBO and El Niño. They found that the stratosphere cannot be neglected in the Earth's angular momentum. Around 20% of the LOD relative to the atmosphere below 100 hPa, belongs to the impact of the stratosphere. Furthermore, they mentioned a small lag (10 - 20 days) between the LOD and variations in the angular momentum, but the lag does not appear to be statistically significant. Therefore only comparisons on seasonal and longer time
scales are useful to be considered. All parameters which are displayed in Figure 6 show a seasonal pattern. First we describe the results for the NH stations. For the NH the zonal wind and the LOD shows decreasing values during summer and increasing values during winter. Beside the striking seasonality, short time fluctuations within a year are observable during the winter in the zonal wind for some years. During the winter of 2010 and 2011, and on even shorter time scales as few months during the winter 2006, 2014 and 2015, decreases in the LOD together with decreases in the zonal wind are visible. The LOD varies
between -1 and 4 milliseconds. The LOD oscillation shows seasonal variations of a fluctuation with shorter day lengths during NH summer and longer day lengths during winter, which fits to the density increase and decrease of the MLT as described above. For the midlatitude station the oscillation pattern in the LOD and the wind are qualitatively similar, but shifted in time. The wind peaks occur earlier in the year than the LOD peaks, which goes along with the earlier wind transition at midlatitudes, which can be seen in Figure 2. For Davis a time shift of approximately half a year occurs between the zonal wind and the LOD,
due to the opposite seasonal wind pattern.

In the summer wind transition altitude a time shift occurs between both parameters. The altitude of the wind transition in these cases is defined as the height between the above located eastward and the below located westward wind during summer. At these heights the wind and the LOD are almost uncorrelated. Above the summer wind transition altitude the oscillation pattern between the LOD and the winds are quite opposite than for 80 km altitude, with a 180° shift between both parameters, which
can be seen in Figure 7. The phase shift, which is pronounced during the summer, obviously results from the opposite wind regime compared to the 80 km altitude. Nevertheless, above the transition height, changes in the density, due to the intensity of the solar radiation, are more pronounced than at lower heights. Therefore the existing seasonal wind pattern fits well to the atmospheric density increase and decrease at these layers.

Additionally, we show in Table 1 correlation coefficients for the 4 locations for the altitudes between 80 and 98 km. Positive
correlation values correspond to the occurrence of an eastward directed wind together with an increased LOD. The values of the NH follow a similar pattern, with positive coefficients below the vertical transition height and negative above. Davis shows a different pattern, with overall negative correlation coefficients. This is owing to the opposite zonal wind pattern compared to the NH. Theoretical, a time shift of $\sim$ half a year would lead to a similar correlation pattern as in the NH.

Figure 8 shows, the mean zonal wind at $\sim$80 km geometric height, based on MLS data, and the LOD. These mean zonal winds
include wind values within the longitude grid between 0°E and 20°E, which is comparable to the NH stations. The Figure is divided in 10° latitude steps centered at latitudes from 80° to 10°S/N. Each latitude grid includes values for +/- 6°. For the MLS observations the comparison between the wind and the LOD are similar to the 80 km meteor results at the respective latitudes. Furthermore, the occurrence of half a year time shift between both polar hemispheres can be seen. A 180° phase shift would lead to the wind-LOD pattern of the opposite hemisphere. Furthermore, the strongest correlation between both parameters can
be seen at northern polar latitudes. Due to an increase in the difference between the geometric and geopotential heights, we

do not show comparisons for higher altitudes. We added correlation coefficients (black) between the mean zonal wind and the LOD for each latitude. A correlation increase towards the northern high latitudes is visible. The same would be seen if a 180° phase shift is added to the time series. Additionally, we present global correlations (green) by averaging mean zonal wind data over all longitudes, whereby possible stationary planetary waves are filtered. The global correlation coefficients are nearly

5 similar to the values for previous average winds between 0-20°E. The shape of the curves between the global average winds are also nearly equal, therefore we didn't add them in the Figure.

In Figures 9 and 10 long term changes of annual LOD (black) and annual mean zonal winds (red) are shown for Collm and for Davis. At this point, we have to mention that the tendency over a long time series is not linear in time. Parameter which influence the tendency of the wind and the LOD also vary over time. Such changes are often approximated by a piecewise

linear trend model (e.g., Tomé and Miranda (2004), Merzlyakov et al. (2009) and Jacobi et al. (2011)), where different linear fit tendencies are estimated for different time periods. Nevertheless, due to the length of the available data series we decided not to use a piecewise linear trend model. The wind values exclude seasonal and solar cycle variations and the LOD excludes the seasonal variations. Exemplary for the location of Collm (Figure 9) the altitudes between 80 and 96 km are displayed. The error bars correspond to the annual variance for each height and the dotted lines show the long term tendency for each

parameter. Figure 9 shows that a long term increase of the LOD occurs together with a long term decrease of the zonal wind. Above 94 km the tendency reverses into a slightly positive wind. This reversal can be explained by the stronger influence due to gravity wave filtering, which has to be considered and cannot be excluded by filtering the data. The tendencies of an increased value for the LOD and a decreased value for the mean zonal wind can be seen for all mid latitude locations, and also for Davis (see Figure 10). Andenes shows for all altitudes increase tendency in the zonal wind (not shown). The results indicate that the

connection between the LOD and the wind is more pronounced at lower latitudes, which is simply explainable by the rotation velocity, which is higher at the middle latitude stations than at the polar latitudes like at Andenes and Davis. The results of an increase of the LOD and a decrease of zonal wind agrees with the relation between fluctuations in the neutral density and the zonal wind, as shown in Stober et al. (2012).

## 4    Conclusion

Within this work we show that the mesospheric winds are affected by an expansion/shrinking of the upper atmosphere that takes place due to changes in the intensity of the solar radiation, which effects the density within the atmosphere. A reason, besides the solar cycle effect, is the annual movement of the Earth around the Sun, which leads to a shorter distance between both celestial bodies during the NH winter, and a longer distance during summer. This leads to a shrinking/expansion of the atmosphere during the NH summer/winter. This shrinking effect takes mainly place in the upper atmosphere, where the amount

of mass is small enough to be sensitive enough to changes to the intensity of solar radiation, as well as temperature changes. According to Stober et al. (2012) an increase of the neutral density together with a decrease of the zonal wind in the MLT region occurs. Based on these findings we showed that a theoretical density increase by 1% between 70 and 100 km leads to a decrease in the atmospheric rotation speed, within a defined layer, by up to 4 m/s. The influence of the Earth-Sun distance on the wind

speed was further investigated using winds from four stations in total, whereby two stations are located at similar high latitudes for the northern and southern hemispheres. The other two meteor radar systems are located at the northern midlatitudes. Based on summer and winter mean wind, we found that during the perihelion, where the MLT expands, a decrease in the zonal wind speed for the respective location occurs together with an increase in the LOD. During the opposite aphelion, an increase in the zonal wind occurs beside a decrease in the day length.

Further, we showed that even after removing the seasonal and the 11 year solar cycle variations the zonal wind and the LOD (fluctuations in the length of a day) are connected. We showed the annual tendency evolution over the whole time period, with the result of an increasing LOD trend together with and more pronounced westward directed wind tendency for the middle latitude stations. This effect weakens at the polar station, which is on the one hand due to a smaller radius, which effects the rotation speed of the atmospheric layer. On the other hand, there are further natural factors, as e.g., the gravity wave drag, who strongly influence these tendencies. Further, we were only able to investigate the connection between these parameters on time scales which are at least one year. On shorter timescales a connection between the LOD and the winds cannot be figured out, the LOD consists of oscillations with at least half a year period and with the currently available data we are not able to fully resolve the superpositions of both parameters. Future work remains necessary to fully understand these effects when global density data measurements are available. Additionally, in future work the estimation of a time lag between the LOD and the winds needs to be considered.

We want to mention that based on our findings a connection between the zonal wind and the LOD exists, which we explain by the variation of the available atmospheric density. Furthermore, we only compare global LOD data with local measurements, and within the MLT exists stronger geophysical effects which drives the wind regime at these altitudes. Within this work we only want to point out this effect, and for closer investigations we need global longtime density data.

*Data availability.*

The Andenes/Juliusruh radar data are available upon request from Gunter Stober (stober@iap-kborn.de).

The Collm radar data are available upon request from Christoph Jacobi (jacobi@rz.uni-leipzig.de).

The Davis radar data are available upon request from Damian Murphy (Damian.Murphy@aad.gov.au).

The Microwave limb sounder data are available at https://mls.jpl.nasa.gov/.

*Authors contributions.*

Sven Wilhelm wrote the manuscript with input from all authors. Furthermore, all co-authors contributed to the data interpretation. Gunter Stober provided the high resolution meteor wind data analysis for all stations and ensured the operation of the Andenes and Juliusruh meteor radar. Vivien Matthias provided the used wind analysis for the mircowave limb sounder data. Christoph Jacobi ensured the operation of the Collm meteor radar and Damian Murphy the Davis meteor radar.

*Competing interests.*

The authors declare that they have no conflict of interest. C. Jacobi is one of the Editors-in-Chief of Annales Geophysicae.

*Acknowledgements.* This work was supported by the WATILA Project (SAW-2015-IAP-1 383). The Operation of the Davis Meteor radar was supported through Australian Antarctic Science projects 2668 and 4025. We thank IERS for providing the used LOD data, which can be found under https://datacenter.iers.org. Furthermore we acknowledge the IAP technicians for the technical support and Jorge L. Chau for discussions at an early stage of the work.

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

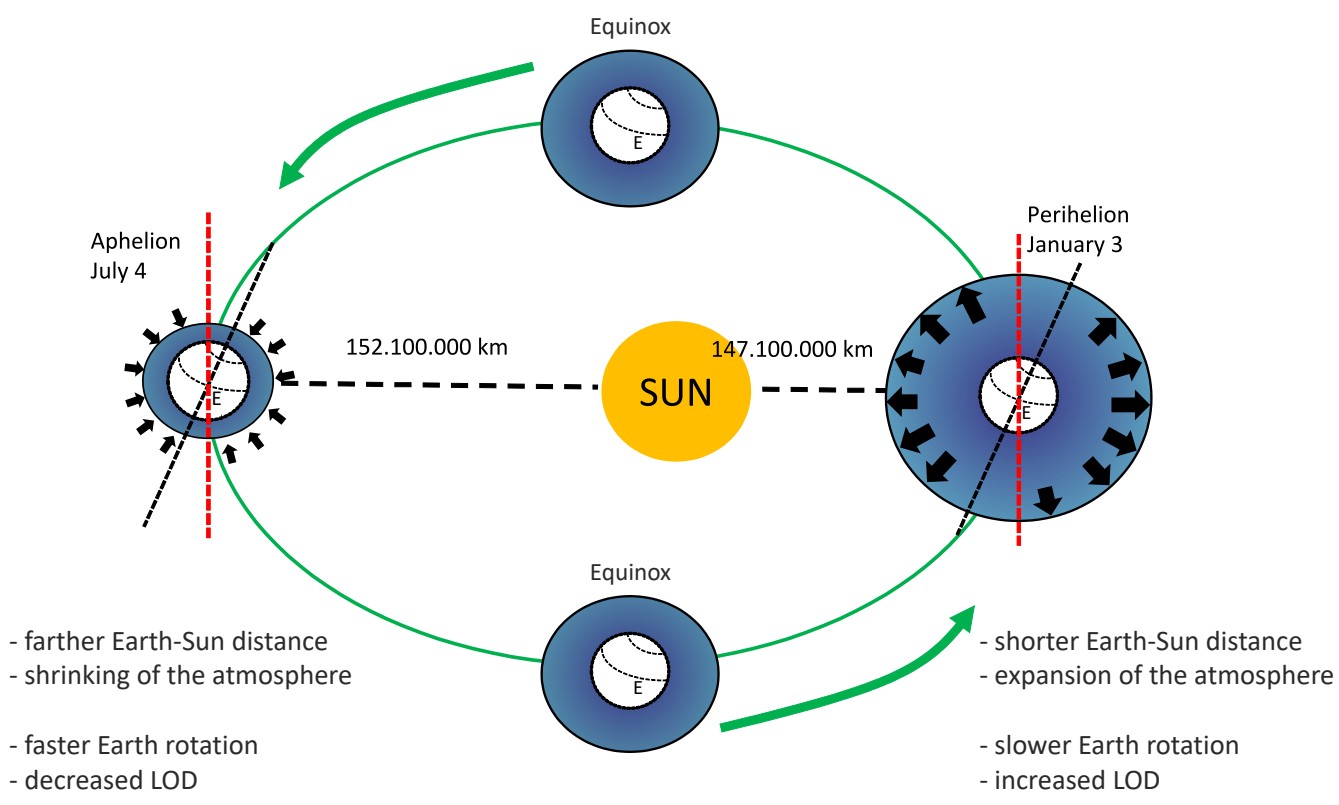

**Figure 1.** Schema of Earth and Sun correlation and the resulting effects on the thickness of the atmosphere and the Earth's rotation velocity.

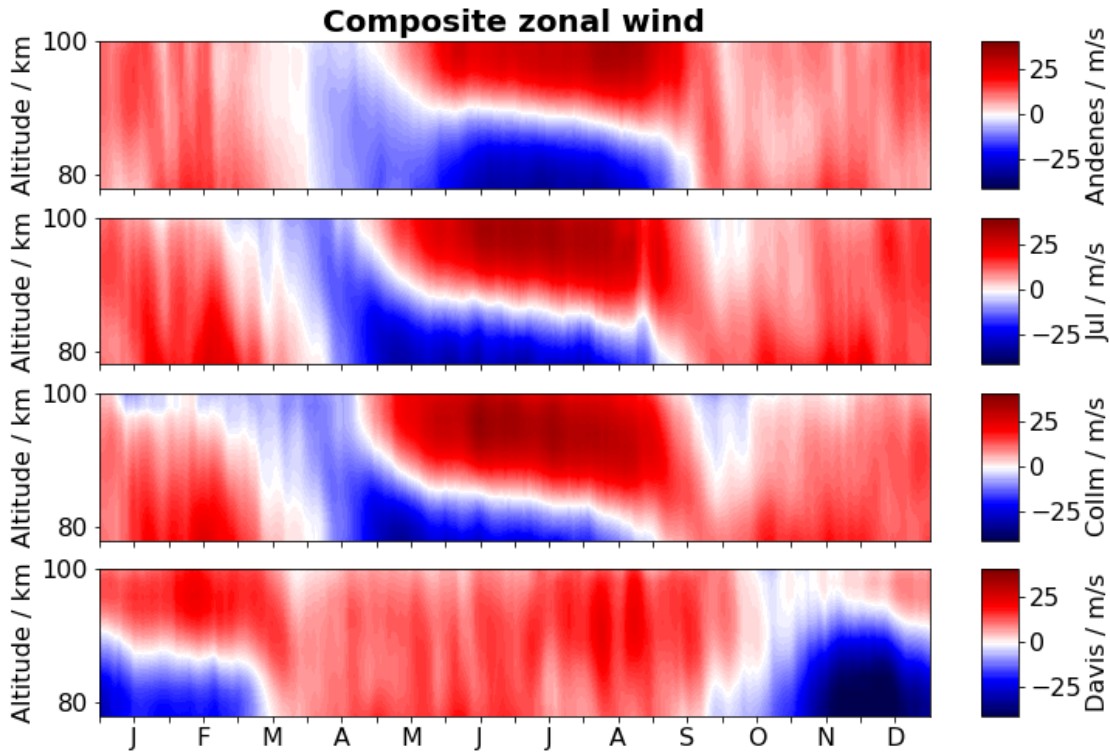

**Figure 2.** Composites of zonal wind for the northern hemisphere stations Andenes (top), Juliusruh (2nd row), and Collm (3th row). At the bottom is shown the southern hemispheric station of Davis. The composite for Andenes, Collm, and Davis include 12 years of meteor radar data and that of Juliusruh 9 years. Positive values correspond to eastward directed winds and negative to westward directed winds.

| km | 80 | 82 | 84 | 86 | 88 | 90 | 92 | 94 | 96 | 98 |
|---|---|---|---|---|---|---|---|---|---|---|
| Andenes | 0.57 | 0.56 | 0.52 | 0.42 | 0.21 | -0.13 | -0.45 | -0.61 | -0.67 | -0.69 |
| Juliusruh | 0.43 | 0.36 | 0.23 | 0.04 | -0.23 | -0.48 | -0.62 | -0.67 | -0.68 | -0.68 |
| Collm | 0.3 | 0.19 | -0.01 | -0.3 | -0.54 | -0.65 | -0.68 | -0.68 | -0.66 | -0.64 |
| Davis | -0.37 | -0.37 | -0.38 | -0.39 | -0.41 | -0.42 | -0.41 | -0.38 | -0.35 | -0.32 |

**Table 1.** Correlation coefficients between the zonal wind and the LOD. Positive values corresponds to the occurrence of e.g., an eastward directed mean zonal wind together with a positive fluctuation in the LOD.

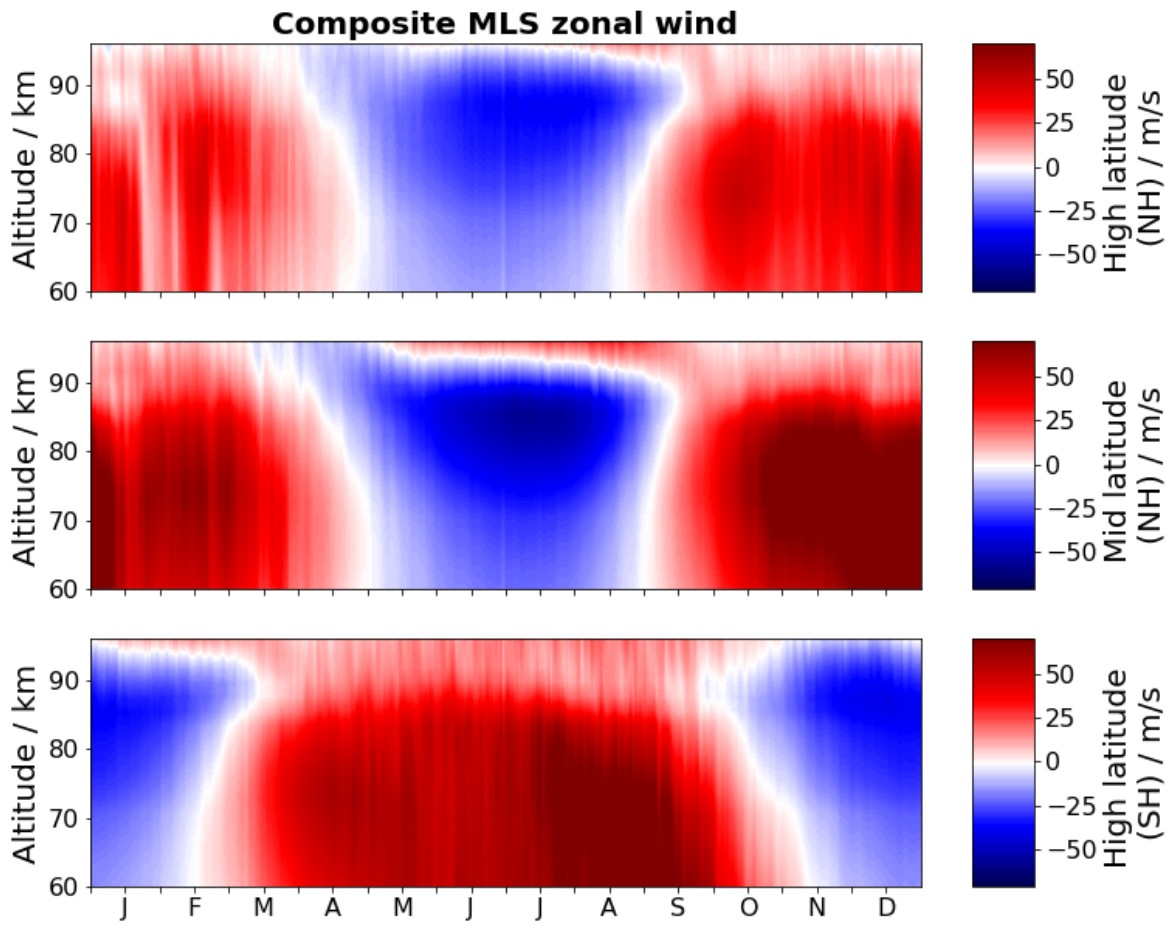

**Figure 3.** Composite of zonal wind for high latitude location (top), and mid latitude location (bottom). The composite of both figures includes 12 years of data wind data derived from MLS geopotential height data. Positive values corresponds to eastward directed winds and negative to westward directed winds. The altitude is given in geopotential height.

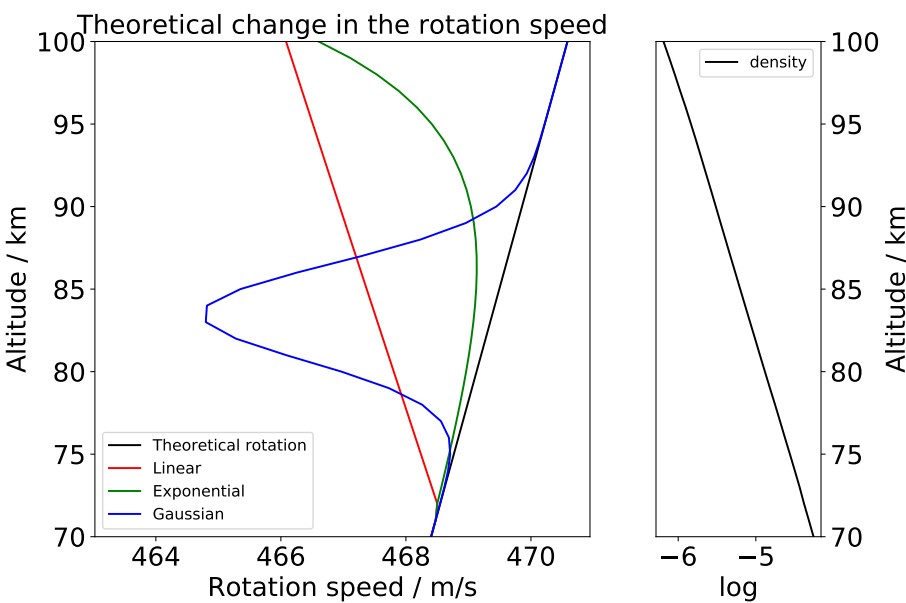

**Figure 4.** Theoretical change of the rotation speed (left side) for a rigid atmospheric layer. In black the theoretical rotation speed of the Earth's atmosphere and in colors the change due to density increase of 1% according the legend. On the right side the density progress is shown for specific altitudes.

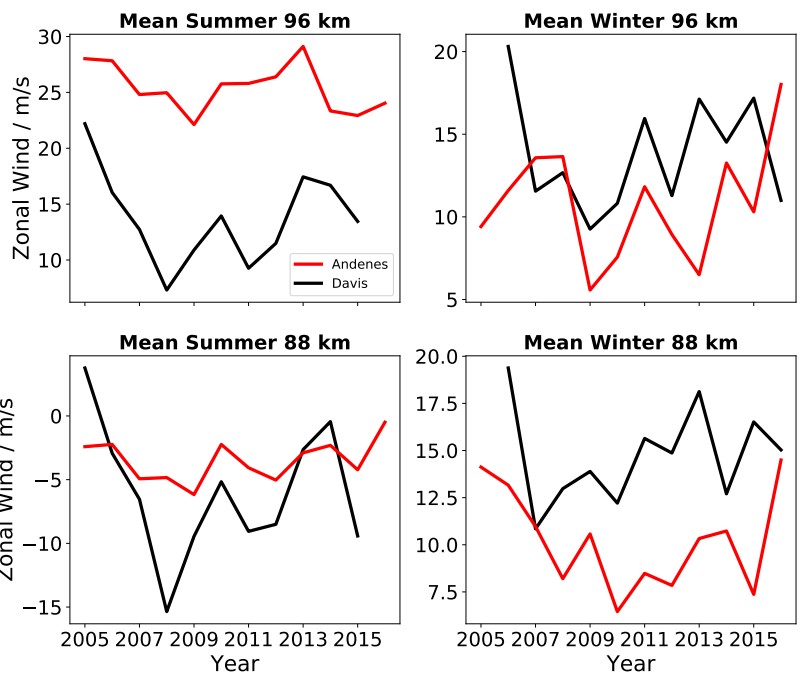

**Figure 5.** Zonal wind amplitudes for winter and summer season at 96 km and 88 km for Andenes and Davis.

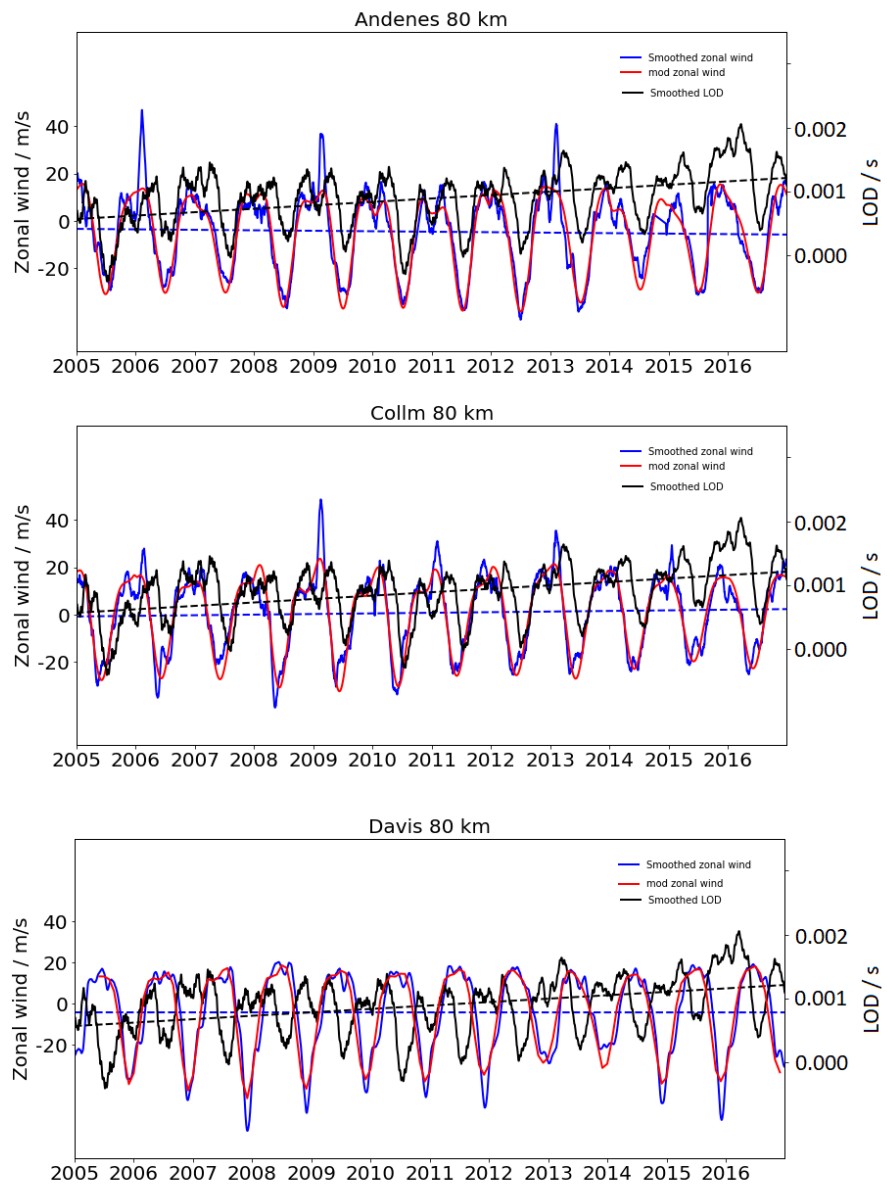

**Figure 6.** Smoothed zonal wind (blue) values based on meteor radar wind data at 80 km and smoothed LOD (black) values. The modulation of the smoothed zonal wind is displayed in red after removing the impact of the solar cycle, whereby the smoothing is stronger as in blue. All curves are done by a smooth over several days, without removing the day-to-day variations, to show the seasonal pattern of the parameters. The dashed lines corresponds to the tendency of the wind/LOD based on linear regression.

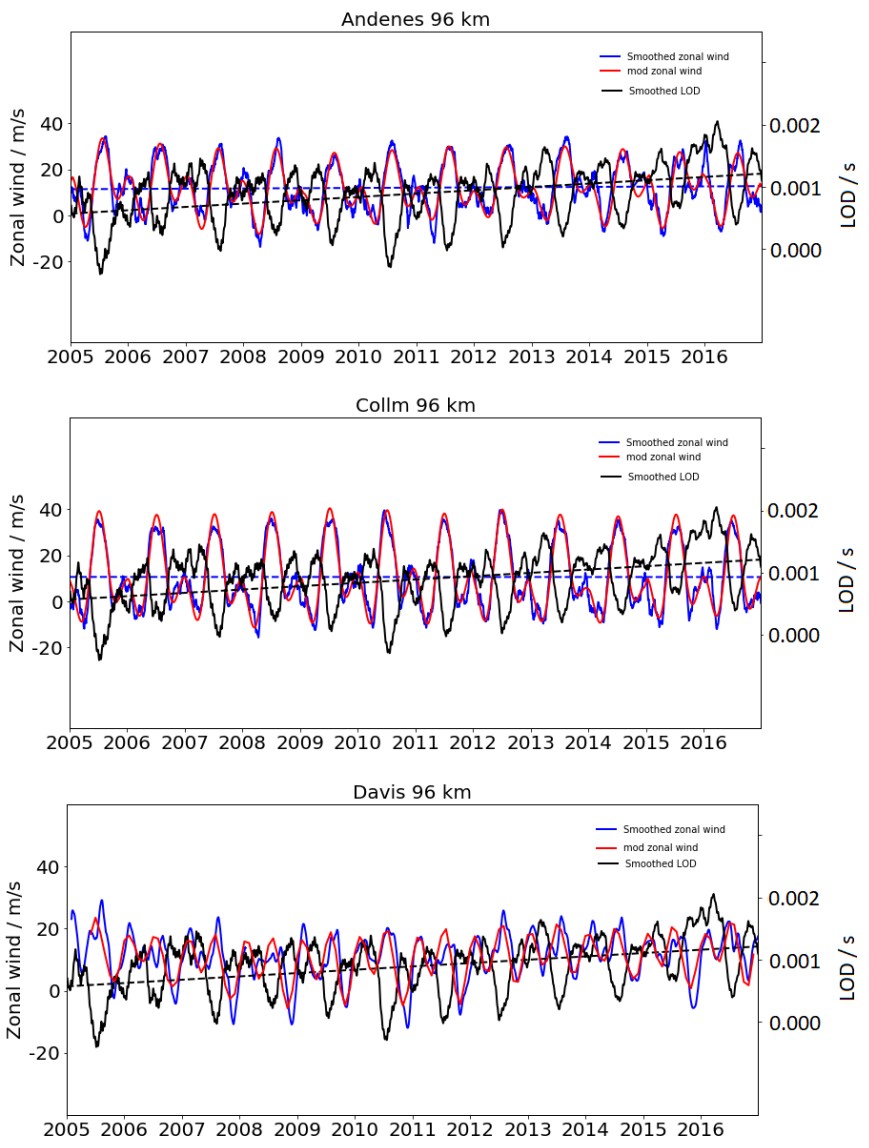

**Figure 7.** Same as Figure 6, but for 96 km.

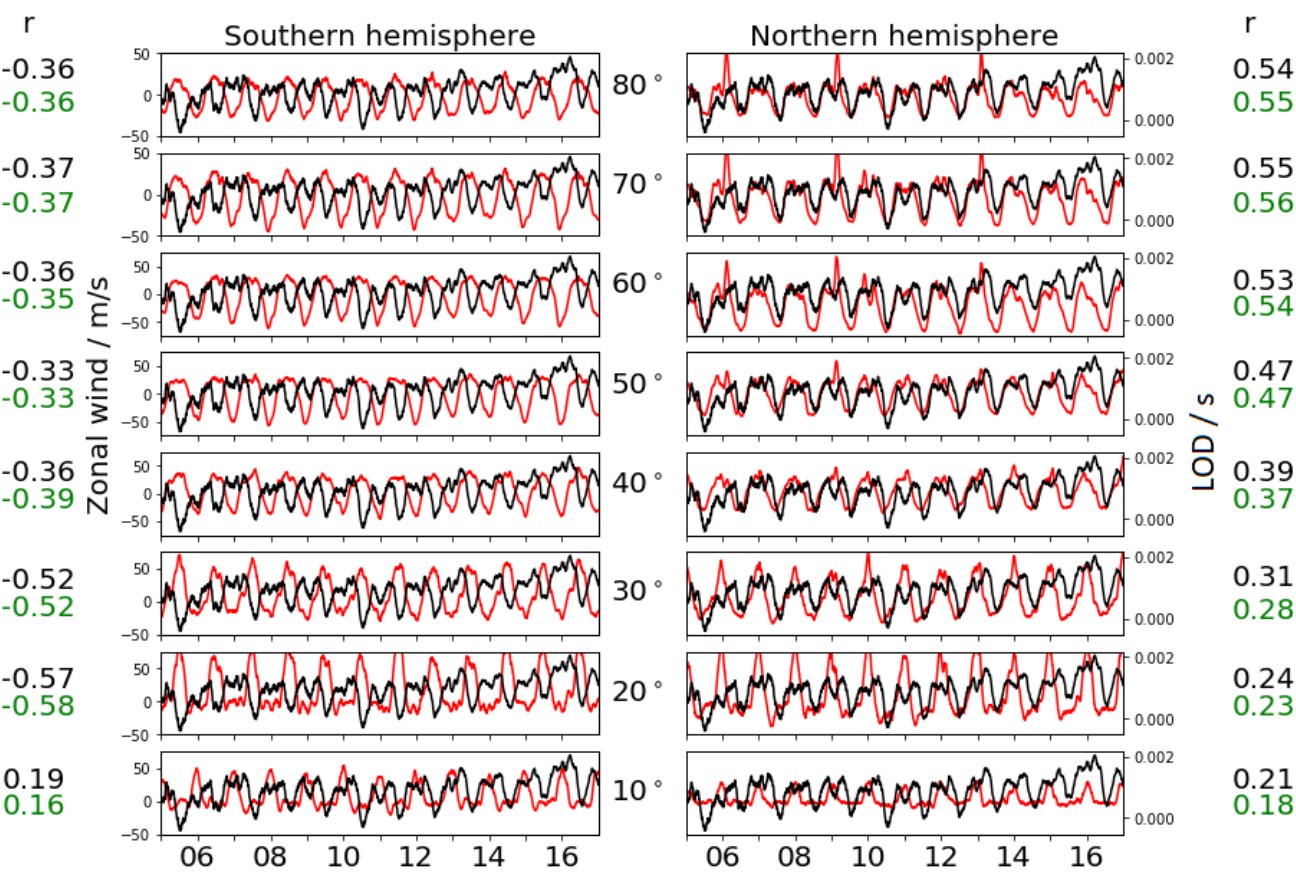

**Figure 8.** Zonal MLS wind (red) and LOD (black) at ∼80 km geometric height for 0°-20°E. The left part show the values for the southern hemisphere, the right for the northern hemisphere, for every 10° latitude. The black correlation coefficients (r) are estimated for the mean between 0°E and 20°E, and the green coefficients corresponds to global average over all longitudes.

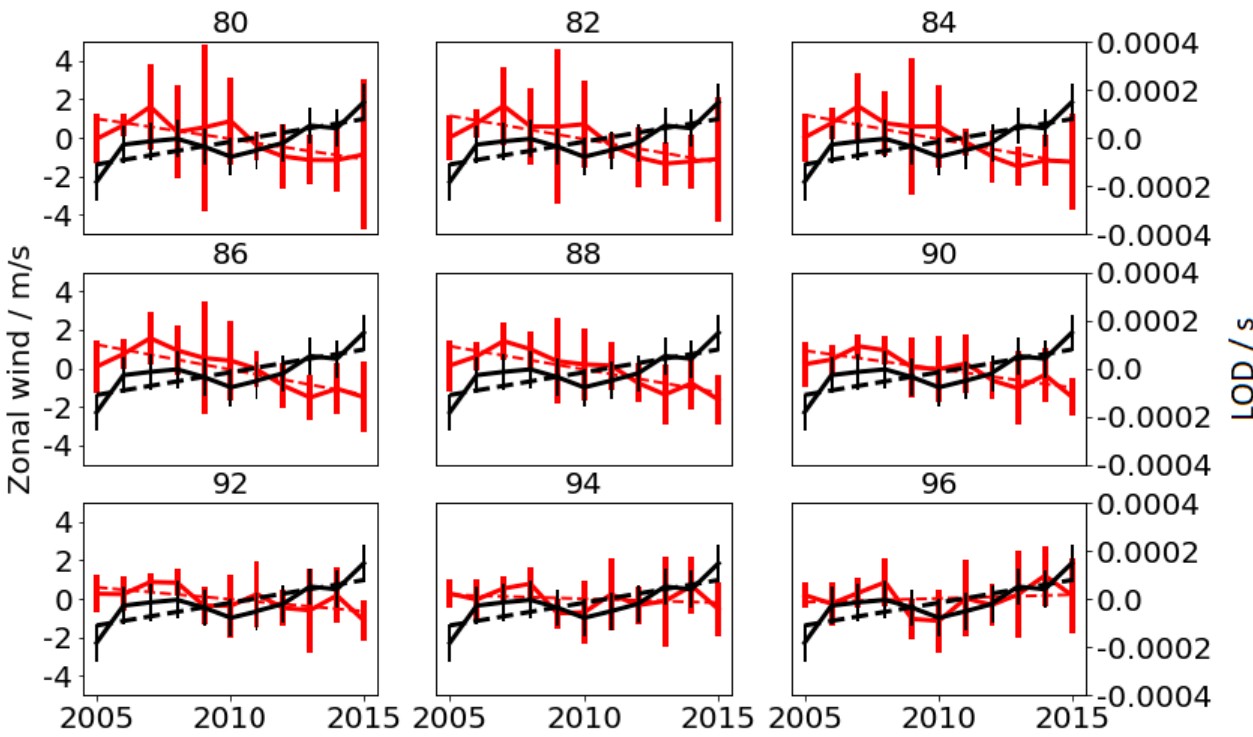

**Figure 9.** Annual mean values for the LOD (black) and the zonal wind (red), for the station Collm, after removing seasonal variations and the solar cycle for the altitudes between 80 and 100 km. The errorbars corresponds to the standard deviation. The dashed lines represents the tendency.

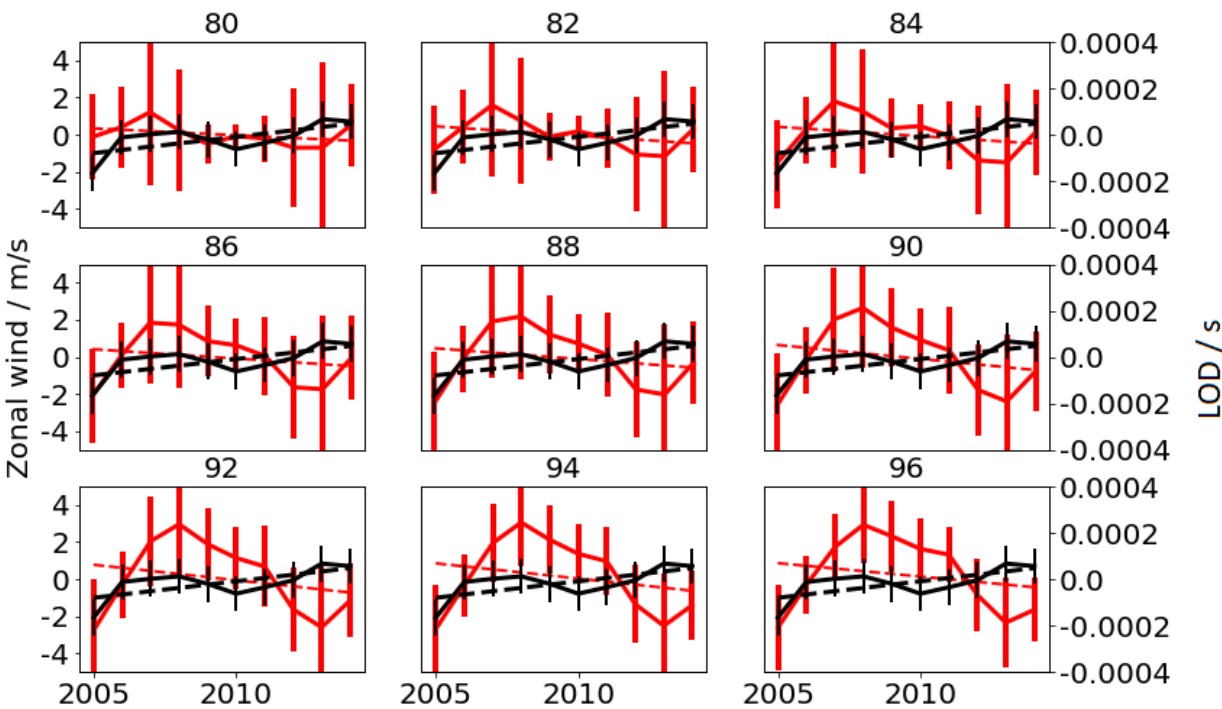

**Figure 10.** Same as Figure 9, but for Davis.