# Peer review of "Connection between the length of day and wind measurements in"

_Annales Geophysicae, 2018_

## Referee Comment (RC1) · Anonymous Referee #1 · 13 Jul 2018

**Review of:** "Connection between the length of day and wind measurements in the meaosphere and lower thermosphere at mid and high latitudes.
by Sven Wilhelm et al. [AnGeo 2018-15, rcvd June 2018]

Relation between zonal wind and length of day variations is a very ambitious topic.

The discussion is very interesting, but I can't find any proof that LOD affects the measured winds.

There are a few points which should be addressed in the introduction.

1. If the earth-atmosphere system were rigid, and the atmosphere expands, the whole system would slow down (become less eastward or more westward) to maintain conservation of angular momentum, But it is not rigid, and since atmospheric drag is the proposed cause of the earth seasonal LOD change, it is important to estimate the time constant. If longer than a season, then maybe no effect on wind would be noticeable. Pg. 8 L 23 mentions the matter, but there is no estimate.

**2.** If the atmosphere is heated why would expansion be simple and not lead to a different climatology, including winds.

**3.** The proposed LOD effect depends on heating: the solar cycle radiation variation is surely bigger than the earth-sun distance effect. Does zonal mean wind show a solar cycle variation?

**4.** It appears that heating to the winter atmosphere should be smaller, even with a closer sun-earth distance, else why is it winter. Why is not atmospheric expansion smaller than in summer?

**Fig. 6,7** seem to be showing the full zonal wind vs. LOD. According to an earlier statement, 4 m/s is the estimated contribution from LOD (e.g. Fig. 4), both having annual variations. How does this figure show the LOD-only contribution?

**Fig. 8** The long term effect of tides and earth deformation are usually taken to be the cause of slowing the earth's rotation, not atmosphere. How does that physically create a trend in the zonal wind.

Minor typos, grammatical, etc.

```
Pg 1  L 10  siderial time
      L 11  full rotation ,   "86400"  to make it international.
            But 86400s is a mean solar day, not a mean siderial day,
            and LOD was said to based on siderial time; the difference
```

is ~ 4 minutes. Some text changes are necessary.
         L 13   deceleration ?

Pg 2  L 10 ``at solar minimum as well as decrease in the temperature  ..." ?
Pg 3  L 22   ``its" ?

Pg 4  L  4   ``... atmosphere were vertically ..."
      L 19   1960s and 70s
      L 23   60s
      L 24   What is "d" ?

Pg 5  L  6   describe
      L  8   `` ... under the assumption of equal density ..."
      L 30   The Aura MLS GPH at 0.001 hPa is virtually always ~90 km
      L 32   `` density-dependent " ?

Pg 6  L  1   ``height and temperature ..", ``horizontal grids which are ..."
      L 23    ~ 90 km  again
      L 24  ``... in qualitatively good agreement ..."

Pg 7  L  2  Geopotential?  or is geometric calculated from geopotential ?
      L 9,12  `` Gaussian"
      L 13  `` these results agree with the observations ..." ?
      L 15  misplaced ``(``
      L 31  these ,   this phenomenon  or these phenomena

Pg 8  L  8  explicitly
      L 22  relatively
      L 34  "solar cycle variation " (there is only one cycle here)

Pg 9  L 14-20 need to be re-written. Meaning is not clear
      L 21 "an overall ..."
      L 22 " is more ..."
      L 33 "stations"

Pg 10 L 10  "can not be figured ..."
      L 16 "...  at these altitudes. "

Pg 11  L35 "using"

Caption Figure 2, 3: "positive"

—///—

---

## Referee Comment (RC2) · Anonymous Referee #2 · 23 Jul 2018

**Review of "Connection between the length of day and wind measurements in the mesosphere and lower thermosphere at mid and high latitudes"**

**General comments**

A topic of this manuscript, study of correlation between a length of a day and zonal mean winds, is very interesting. However, I concern about three points.

Although data analysis results are presented in 6 figures in the manuscript, all of them are about mean zonal winds from meteor radars. Among them, only 3 figures overplot lengths of a day. Trends of lengths of a day and zonal mean winds look somehow correlated in the figures, but correlations are not presented numerically. Because mean winds are presented in terms of seasonal and interannual variations. I would like to have seen them for lengths of a day, too.

Zonal mean winds are presented using meteor radar measurements at 3 sites in the northern hemisphere and the high latitude southern hemisphere, and Aura/MLS. I expected that Aura/MLS data results are compared with radar results for a validation and then would present any global variations. However, Aura/MLS results are used only for comparisons of radar results in the northern hemisphere. Because authors conclude zonal mean winds agree between MLS/Aura and radar measurements, I do not understand motivation of presenting trends of lengths of a day and zonal mean winds from both meteor radars (Figure 6) and Aura/MLS (Figure 7).

Lastly, authors compare zonal mean winds using Andenes and Davis in the northern and southern hemisphere in a same season and conclude that a difference is caused only by lengths of a day between northern hemisphere summer and winter. However, I believe that mean winds from ground measurement only one site include zonal mean wind and stationary planetary waves and difference of stationary planetary wave amplitudes largely results to a difference of mean winds of ground measurements. I also believe that main reason of interhemispheric differences in atmospheric dynamics is a difference of topography. It makes a difference of atmospheric waves with interact with mean winds. The difference of topography makes an interhemispheric difference of chemical composition in the atmosphere, such as water vapor, ozone, and carbon dioxide, which makes an interhemispheric difference of viscosity and then winds.

I report that this manuscript needs further consideration and discussion.

**Specific comments**

Line 24 on page 4: What is $d$? Because equation (7) shows $d(t)$, it must be a variable parameter depending on time (I expect that $t$ stands for time). Can authors change "$d$" to another symbol or acronym because it is very confusing with integral and differential symbols?

Lines 21 to 24 on page 5: I am suspicious if you can estimate winds at 78 (or below ~85 km) and 100 km by meteor radars. What is an altitude resolution, and every how much in km did authors determine hourly mean winds? Is there any threshold for a determination, such as elevation angle, range, minimum and maximum radial velocities, and minimum number of sampling meteor echoes? Although authors mention uncertainties as "between 2 and 6 m/s", weightings of uncertainties are very different between 50 m/s wind with 6 m/s uncertainty and 5 m/s with 6 m/s uncertainty.

Line 31 on page 5: Please check a vertical resolution. In my knowledge, Aura/MLS data are every 1.3 km up to 50 km, 2.7 km up to 62 km and 5.4 km above.

Line 18 on page 6: Juliusruh and Collm are at nearly same location in a global sense. What causes a difference of reversal altitudes by 3 km? Are they systematic difference?

Lines 9 to 10 on page 8: How was "the fluctuation in the LOD" obtained? Was it by equation (7)? If so, what is $d(t)$, as asked above? Was $d(t)$ obtained from measurements or some simulation models?

Line 22 on page 8: What is "the F10.7 solar cycle"? Is it the 11-year cycle, the 27-day cycle, or both cycles?

Line 29 on page 8: LOD (either length of a day or fluctuation in a length of a day) must have unit of time (probably second from Figures 6 to 8). Why is an LOD unit ms (millisecond or meter times second)?

Line 33 on page 8: Again, please make sure what "the solar cycle" is, 11 year, 27 day, both, or some other cycle? Also, how much does "the solar cycle" influence on a fluctuation of a length of a day? It means how much important to remove a solar cycle influence.

Line 12 on page 9: What is "the size range"?

Line 25 on page 9: I do not see that the authors showed affects on mesospheric winds by expansion/shrinking of the upper atmosphere in this work. I saw that the authors showed correlations between zonal winds in the mesosphere and fluctuations in a length of a day. Stober et al. inferred that fluctuations in a length of a day are correlated with a variation of a thickness of the upper atmosphere. It is possible to expect that zonal winds in the mesosphere are related to a thickness of the upper atmosphere. Please revise it.

Figure 2 and 3: Why do they exclude Davis in Figure 2 and southern hemisphere?

Figure 2 and 3 captions: Correct to "positive".

Figure 6: Please describe what black and blue dashed lines are in a caption.

**Technical corrections**

Line 10 on page 1: Do authors use LOD as "length of a day" or "fluctuations in a length of a day" in this manuscript? Line 24 on page 4 says "length of day (LOD)". However, "LOD" is used in a subsection 3.2 and a caption of Figure 6 although most of them must imply "fluctuation of LOD", except for lines from 9 to 10 on page 8 say "fluctuation in the LOD". Please make it consistent.

Lines 9 to 10 on page 2: "shrinking of the middle atmosphere between solar minimum and solar maximum" is very confusing. Does the middle atmosphere shrink at the solar minimum, the solar maximum, or both at the solar minimum and maximum?

Line 19 on page 2: Does "conversation" mean "conservation"?

Line 8 on page 5: What does "on an in average" mean?

Line 4 on page 6: I feel that "combined 04 data from the international Earth Rotation and Reference System Service (IERS)" is more appropriate.

Line 24 on page 6: "qualitatively"?

Lines 24 to 25 on page 6: I do not understand the sentence and suggest revision.

Line 30 on page 6: Are MLS data shown in a geopotential height? If so, why "above 90 km" is suddenly described in geometric height?

Lines 6 to 7 on page 7: It is very ambiguous. Does a density increase occur in summer OR winter, and at the solar minimum OR maximum?

Line 16 on page 7: Change "the northern and the southern hemisphere" to "the northern and southern hemispheres".

Line 16 on page 7: Remove comma between "opposite" and "fluctuations".

Lines 20 to 21 on page 7: What is "between two locations on the same latitude"? Does it mean "at the same latitude in the northern and southern hemispheres"?

Line 21 on page 8: It should be "additionally".

Line 22 on page 8: It should be "relatively".

Line 29 on page 8: Please make sure if "seasonal fluctuation" means "seasonal variation of a fluctuation", "seasonal means of fluctuation", or something else.

Line 33 on page 8: What does "as result as" mean?

Line 19 on page 9: "This reversal can be explain can be explain" must be "This reversal can be explained".

Lines 20 to 21 on page 9: First, "station" on line 21 must be "stations"? What are "the polar and the second midlatitude stations"? "the polar stations" include both Andenes and Davis? Is "the second milatitude station (I think not "stations" in this case)" Juliusruh or Collm? Did the authors define "first" and "second" stations previously?

Line 1 on page 10: "hemisphere" must be "hemispheres".

Lines 6 to 7 on page 10: Why do authors specify "the middle latitude stations" as Collm and Juliusruh? Is "the polar station" only Andenes? How about Davis?

Line 10 on page 10: "not figured out" must be "not be figured out".

Line 13 to 14: I do not understand this sentence. Please revise it.

Line 1 on page 11: "ssignal" must be "signal".

Line 20 on page 11: "datadata" must be "data".

---

## Author Comment (AC1) · 5 Sep 2018

Review of: "Connection between the length of day and wind measurements in the mesosphere and lower thermosphere at mid and high latitudes." by Sven Wilhelm et al. [AnGeo 2018-15, rcvd June 2018]

Relation between zonal wind and length of day variations is a very ambitious topic. The discussion is very interesting, but I can't find any proof that LOD affects the measured winds.

General reply: We thank the Referee for this constructive suggestions and comments

that help to improve the paper. In the supplement you can find the corrected version of the manuscript.

Aim of this study is to show that, according to previous work by Stober et al. (2012), the mesospheric/lower thermospheric zonal wind is connected to changes in the neutral density. Due to a lack of global density observations these concern are hard to proof. Therefore we took the LOD for comparison to the zonal wind, because the LOD is related to changes in the global Earth rotation speed. These changes can be influenced by large scale atmospheric processes and we further state that within the MLT the atmospheric rotation speed is also affected by changes in the neutral density. We figured out that the connection occurs on annual or longer time scales, and further follows the influence of solar radiation (on 11-year solar cycle scale, as well as differences in the Earth-Sun-distance). On shorter time scales, only based on wind measurements, a connection between the LOD and MLT winds, with respect to the impact of the Sun are quite complicated to find. We will leave a true understanding of the effect, which implies exact velocity values, for future work.

There are a few points which should be addressed in the introduction.

1. If the earth-atmosphere system were rigid, and the atmosphere expands, the whole system would slow down (become less eastward or more westward) to maintain conservation of angular momentum, But it is not rigid, and since atmospheric drag is the proposed cause of the earth seasonal LOD change, it is important to estimate the time constant. If longer than a season, then maybe no effect on wind would be noticeable. Pg. 8 L 23 mentions the matter, but there is no estimate.

Reply: We agree that a calculation of a time constant is important to estimate a direct LOD effect on higher located winds, but only based on our available radar measurements we are not able to estimate the lag. We will mention this for future work. Nevertheless, we added the following part to the paper:

According to e.g., Dickey et al. (1994) a direct effect exists between the stratospheric
and tropospheric zonal wind and the LOD on annual time scales due to long term geophysical effects, as e.g., QBO and El Nino. They found that the stratosphere cannot be neglected in the Earth's angular momentum. Around 20 % of the LOD relative to the atmosphere below 100 mbar, belongs to the impact of the stratosphere. Furthermore, they mentioned a small lag (10 - 20 days) between the LOD and variations in the angular momentum, but the lag do not appear to be statistically significant.

Dickey, J., O., Marcus, S., L., Hide, R., Eubanks, T., M., and Boggs, D., H.: Angular momentum exchange among the solid Earth, atmosphere, and oceans: A case study of the 1982 - 1983 El Nino event., Journal of geophysical research, 99, 23, 921– 23,937, 1994.

2. If the atmosphere is heated why would expansion be simple and not lead to a different climatology, including winds. Reply: We assume that the expansion/shrinking of the atmosphere also influence the climatology, but the effect is relatively small compared to the atmospheric drivers, as gravity waves and chemistry. In a model simulation Marsh et al. (2007) showed for the whole atmosphere a response to changes in the 11-year solar cycle, with e.g., the result of temperature changes in the lower thermosphere by over 100 K at solar maximum relative to solar minimum. Further they showed the occurrence of tropospheric wind and temperature changes due to changes in the solar radiation. But they also mention that changes in the climatology due to solar radiation are too complex to lead to simplified results. Furthermore, there are other factors as e.g., the composition of chemical components and the occurrence and propagation of gravity waves which lead to the state of the climatology.

We added/reformulated some text in the manuscript to clarify the point of the study:

We have to note that beside many others factors, this is only one reason, and by far not the dominant factor, for the wind differences between both locations at theses altitudes. Other physical processes have also a strong effect on the hemispheric wind differences e.g., the topography, chemical composition of the atmosphere (Marsh (2007),

Lee (2018)), and the occurrence and propagation of gravity waves. These waves are the main drivers of the atmospheric wind circulation and therefore also influence the local wind differences at both hemispheres. Furthermore gravity waves lead, compared to the annual mean, to a colder summer mesosphere and a warmer winter mesosphere e.g., Luebken (2014). These temperature differences also fit well to the atmospheric expansion/shrinking. Unfortunately, based only on wind measurements we are not able to estimate a precise value on how strong the connection is between zonal mean wind with the LOD. For a more detailed understanding of these phenomena global density observations would be required.

and

In a model simulation Marsh et al. (2007) showed for the whole atmosphere a responds to changes in the 11-year solar cycle, with e.g., the result of temperature changes in the lower thermosphere by over 100 K at solar maximum relative to solar minimum. Further they showed the occurrence of tropospheric wind and temperature changes due to changes in the solar radiation. But they also mention that changes in the climatology due to solar radiation are too complex to lead to simplified results.

Lübken, F.-J., Höffner, J., Kaifler, B., and Morris, R., J.: Winter/summer mesopause temperature transition at Davis (69°S) in 2011/2012, Geophys. Res. Lett., 41, 5233–5238, https://doi.org/10.1002/2014GL060777, 2014.

Marsh, D., R., Garcia, R., R., Kinnison, D., E., Boville, B., A., Sassi, F., Solomon, S., C., and Matthes, K.: Modeling the whole atmosphere response to solar cycle changes in radiative and geomagnetic forcing, Journal of geophysical research, 112, https://doi.org/doi:10.1029/2006JD008306, 2007.

Lee, J., N., Wu, D., L. R. A., and Fontenla, J.: Solar cycle variations in mesopheric carbon monoxide, Journal of atmospheric and solarterrestial physics, 170, 21–34, https://doi.org/https://doi.org/10.1016/j.jastp.2018.02.001, 2018.

3. The proposed LOD effect depends on heating: the solar cycle radiation variation is surely bigger than the earth-sun distance effect. Does zonal mean wind show a solar cycle variation?

Reply: Yes, the zonal mean wind shows a solar cycle variation. We added a Figure, where for the location of Andenes the zonal mean wind between 84 and 94 km is displayed together with the F10.7 solar cycle index. As result is shown that for a low F10.7 index an enhanced westward wind appears, while for stronger a F10.7 index the eastward directed wind gets enhanced. Further also a shift in the summer vertical wind shear occurs, which is also correlated to the solar cycle. There occurs a shift onto higher altitudes together with a decrease of the solar radiation, due to a change in the neutral density. This pattern can also be seen for the other locations.

We added the text: To underline this statement, Figure 4 shows, for the location of Andenes, the zonal mean wind between 84 and 94 km together with the F10.7 11-year solar cycle index (black line). An enhancement of the eastward directed wind occurs together with a stronger F10.7 index and more clearly an increase of the westward directed wind together with a smaller F10.7. Furthermore a shift occurs in the summer vertical wind shear, which is also correlated with the solar cycle, whereby a shift to higher altitudes takes place together with a decrease of the solar radiation, due to a change in the neutral density.

FIGURE 4

Caption (Figure 4.. see attachment): Zonal mean wind for Andenes for the heights between 84 and 94 km, together with the F10.7 11 year solar cycle index in black.

4. It appears that heating to the winter atmosphere should be smaller, even with a closer sun-earth distance, else why is it winter. Why is not atmospheric expansion smaller than in summer?

Reply: Propagation of gravity waves which breaks in the mesosphere leads, compared

to the annual mean/radiative equilibrium, to a cold mesosphere during the summer and a warm mesosphere during the winter. This temperature difference fits well with the atmospheric shrinking/expansion.

We added a comment in the text: (see point 2.)

Other physical processes have also a strong effect on the hemispheric wind differences e.g., the topography, chemical composition of the atmosphere (Marsh (2007), Lee (2018), and the occurrence and propagation of gravity waves. These waves are the main drivers of the atmospheric wind circulation and therefore also influence the local wind differences at both hemispheres. Furthermore gravity waves lead, compared to the annual mean, to a colder summer mesosphere and a warmer winter mesosphere e.g., Luebken (2014). These temperature differences also fit well to the atmospheric expansion/shrinking.

Marsh, D., R., Garcia, R., R., Kinnison, D., E., Boville, B., A., Sassi, F., Solomon, S., C., and Matthes, K.: Modeling the whole atmosphere response to solar cycle changes in radiative and geomagnetic forcing, Journal of geophysical research, 112, https://doi.org/doi:10.1029/2006JD008306, 2007.

Lee, J., N., Wu, D., L. R. A., and Fontenla, J.: Solar cycle variations in mesopheric carbon monoxide, Journal of atmospheric and solarterrestial physics, 170, 21–34, https://doi.org/https://doi.org/10.1016/j.jastp.2018.02.001, 2018.

Lübken, F.-J., J. Höffner, T. P. Viehl, B. Kaifler, and R. J. Morris (2014), Winter/ summer mesopause temperature transition at Davis (69âŮęS) in 2011/2012, Geophys. Res. Lett., 41, 5233–5238, doi:10.1002/2014GL060777.

Fig. 6,7 seem to be showing the full zonal wind vs. LOD. According to an earlier statement, 4 m/s is the estimated contribution from LOD (e.g. Fig. 4), both having annual variations. How does this figure show the LOD-only contribution?

Reply: A precise estimation of the impact of the LOD on the wind is difficult. Based

on the wind observations only we are not able to estimate a correct value for the LOD contribution. According to the equations 1-4 estimations could be done, but due to a lack of density measurements we are not able to determine correct values. This point is added as an outlook for future work.

Fig. 8 The long term effect of tides and earth deformation are usually taken to be the cause of slowing the earth's rotation, not atmosphere. How does that physically create a trend in the zonal wind.

Reply: We added this text to the introduction:

On short time scales a change in the Earth rotation can lead to an uneven heating of the Earth's surface, which results to temperature differences between the surface and the atmosphere above. This can further cause convection currents, which leads to pressure differences in the atmosphere and results in a different wind formation, which can influence the LOD. On longer time scale and especially on higher altitudes increases the importance of the solar influence. An increase of the solar radiation, which can be caused due to a slowing of the Earth's rotation, leads to an expansion of the higher atmosphere, which further results, due to the conversation of angular momentum, in a slower rotation of the atmosphere. What further needs to be considered is e.g., the influence of volcanic eruptions, which influence the Earth's rotation as well as the atmospheric chemistry/temperature (e.g., She et al. (2015)). Changes in these parameters can further lead to changes in the neutral density.

She, C., Krueger, D., A., and Yuan, T.: Long-term midlatitude mesopause region temperature trend deduced from quarter century (1990- 2014) NA lidar observations, Annales Geophysicae, 33, 363–369, https://doi.org/doi:10.5194/angeocom-33-363-2015, 2015.

Minor typos, grammatical, etc.

General Reply: Thanks for the advices. We will correct the mentioned points, and

added here for some points few comments for the Referee.

Pg 1 L 10 siderial time L 11 full rotation , "86400" to make it international. But 86400s is a mean solar day, not a mean siderial day, and LOD was said to based on siderial time; the difference is $\sim$ 4 minutes. Some text changes are necessary.

Reply : we added some text : Within the estimation of the LOD the sidereal time gets converted into solar time, by taking into account the Earth's position and motion with respect to the stars (Aoki, 1981).

Aoki, S., Guinot, B., Kaplan, G., Kinoshita, H., McCarthy, D., and Seidelmann, P.: The new Definition of Universal Time, Astronomy and Astrophysics, 105, 359–361, 1981.

L 13 deceleration ? – no, eastward directed wind (or westerly) leads to an acceleration of the Earth's rotation.

Pg 2 L 10 at solar minimum as well as decrease in the temperature ... ?

Reply: We reformulated the sentence and added some more references:

Previous studies as, e.g., Walterscheid (1989), Marsh et al (2007), Emmert (2015), and Lee et al. (2018) showed that solar cycle variations affects the atmospheric density, temperature, chemical composition and winds over the whole atmosphere, but in particular, in the MTI (Mesosphere-Thermosphere-Ionosphere) system.

Walterscheid, R., L.: Solar Cycle effects on the upper atmosphere: Implications for Satellite Drag, Journal of spacecraft and rockets, 26, 439–444, https://doi.org/DOI: 10.2514/3.26089, 1989.

Marsh, D., R., Garcia, R., R., Kinnison, D., E., Boville, B., A., Sassi, F., Solomon, S., C., and Matthes, K.: Modeling the whole atmosphere response to solar cycle changes in radiative and geomagnetic forcing, Journal of geophysical research, 112, https://doi.org/doi:10.1029/2006JD008306, 2007.

Emmert, J. T.: Altitude and solar activity dependence of 1967-2005 thermospheric density trends derived from obrital drag, Journal of geophysical research: space physics, 120, 2940–2950, https://doi.org/doi:10.1002/2015JA021047., 2015.

Lee, J., N., Wu, D., L. R. A., and Fontenla, J.: Solar cycle variations in mesopheric carbon monoxide, Journal of atmospheric and solar terrestial physics, 170, 21–34, https://doi.org/https://doi.org/10.1016/j.jastp.2018.02.001, 2018.

Pg 3 L 22 its? –corrected-

Pg 4 L 4 "... atmosphere were vertically ..." –corrected- L 19 1960s and 70s –corrected- L 23 60s –corrected- L 24 What is "d" ? We changed d to D, which is the angular velocity of the Earth. To avoid misunderstandings we didn't choose $\omega$, because it is already used in the equations 1,5 and 6 as angular velocity for an altitude defined atmospheric layer.

Pg 5 L 6 describe –corrected- L 8 " ... under the assumption of equal density ..." –corrected- L 30 The Aura MLS GPH at 0.001 hPa is virtually always ∼90 km Reply: We converted the geopotential height into geometric height, as e.g., done in Matthias et al (2013). We added some text to clarify this.

Added text: The geometric heights are approximately estimated from pressure levels as described in Matthias (2013): h = -7 ln(9/1000), where h is the altitude in km and p the pressure in hPa. Furthermore, we are aware about a difference between the geometric and geopotential heights, which increase especially above 80 km. Therefore, we focus in this work on the height range between 60 to 80 km. . .

Matthias, V., Hoffmann, P., Manson, A., Meek, C., Stober, G., Brown, P., and Rapp, M.: The impact of planetary waves on the latitudinal displacement of sudden stratospheric warmings, Ann. Geophys., 31, 1397-1415, https://doi.org/10.5194/angeo-31-1397-2013, 2013.

L 32 " density-dependent " ? –see above

Pg 6 L 1 "height and temperature ..", "horizontal grids which are ..." –corrected- L 23 ∼

90 km again –corrected- L 24 "... in qualitatively good agreement ..." –corrected-

Pg 7 L 2 Geopotential? or is geometric calculated from geopotential ? – we added some more words to clarify this.

L 9,12 " Gaussian" –corrected-

L 13 " these results agree with the observations ..." ? Only based on wind measurements we are not able to extract a specific value.

L 15 misplaced "(" -corrected- L 31 these , this phenomenon or these phenomena –corrected-

Pg 8 L 8 explicitly –corrected- L 22 relatively –corrected- L 34 "solar cycle variation " (there is only one cycle here) –corrected-

Pg 9 L 14-20 need to be re-written. Reply: we reformulated the part:

In the Figures 10 and 11 are shown long term changes of annual LOD (black) and annual zonal mean winds (red) for Collm and for Davis. At this point, we have to mention that a tendency over a long time series is not linear in time. Parameter which influence the tendency of the wind and the LOD also vary over time and therefore be observed in long time series should be limited within a specific period. Such changes are often be approximated by a piecewise linear trend model (e.g., Tomé and Miranda (2004), Merzlyakov et al. (2009) and Jacobi et al. (2011)), where different linear fit tendencies are estimated for different time periods. Nevertheless, due to the length of the available data series we decide to use no piecewise linear trend model. The wind values exclude seasonal and solar cycle variations and the LOD excludes the seasonal variations. Exemplary for the locations of Collm (Figure 10) the altitudes between 80 and 96 km are displayed. The errorbars corresponds to the annual variance for each height and the dotted lines show the long term tendency for each parameter. The result is that a long term increase of the LOD occurs together with a long term decrease of the zonal wind. Above 94 km the tendency reverses for the mid latitude locations into

a slightly positive wind. This reversal can be explain by the stronger influence due to gravity wave filtering, which has to be considered and cannot be excluded by filtering the data. The tendencies of an increased value for the LOD and a decreased value for the zonal mean wind can be seen for all mid latitude locations and also for Davis (see Figure 11). Andenes shows for all altitudes increase tendency in the zonal wind. The results indicates that the connection between the LOD and the wind are more pronounced at lower latitudes, which simply explainable by the rotation velocity, which is higher at the middle latitude stations than at the polar latitudes like Andenes and Davis. The results of an increase of the LOD and a decrease of zonal wind fits to the relation between fluctuations in the neutral density and the zonal wind, as shown Stober et al. (2012).

Tomé, A., R. and Miranda, P., M. A.: Piecewise linear fitting and trend changing points of climate parameters, Geophys. Res. Lett., 31, https://doi.org/doi:10.1029/2003GL019100, 2004.

Merzlyakov, E., G., Jacobi, C., Portnyagin, Yu., I., and Solovjova, T., V.: Structural changes in trend parameters of the MLT winds based on wind measurements at Obninsk (55°N, 37°E) and Collm (52°N, 15°E), Journal of atmospheric and solar-terrestial physics, 71, 1547–1557, https://doi.org/doi:10.1016/j.jastp.2009.05.013, 2009.

Jacobi, C., Hoffmann, P., Liu, R., Q., Merzlyakov, E., G., Portnyagin, Yu., I., Manson, A., H., and Meek, C., E.: Long-term trends, their changes, and interannual variability of Northern Hemisphere midlatitude MLT winds, Journal of Atmospheric and Solar-Terrestrial Physics, 75-76, 81–91, https://doi.org/doi:10.1016/j.jastp.2011.03.016, 2011.

L 21 "an overall ..." –corrected- L 22 " is more ..." –corrected-

L 33 "stations" –corrected-

Pg 10 L 10 "can not be figured ..." –corrected- L 16 "... at these altitudes. " –corrected-

Pg 11 L35 "using" –corrected- Caption Figure 2, 3: "positive" –corrected-

Please also note the supplement to this comment:
https://www.ann-geophys-discuss.net/angeo-2018-51/angeo-2018-51-AC1-supplement.zip

———————————————————

[Figure]

**Fig. 1.**

---

## Author Comment (AC2) · 5 Sep 2018

Review of "Connection between the length of day and wind measurements in the mesosphere and lower thermosphere at mid and high latitudes" General comments

A topic of this manuscript, study of correlation between a length of a day and zonal mean winds, is very interesting. However, I concern about three points.

General reply: We thank the Referee for this constructive suggestions and comments that help to improve the paper. We attached in the supplements a corrected version of the manuscript.

Although data analysis results are presented in 6 figures in the manuscript, all of them are about mean zonal winds from meteor radars. Among them, only 3 figures overplot lengths of a day. Trends of lengths of a day and zonal mean winds look somehow correlated in the figures, but correlations are not presented numerically. Because mean winds are presented in terms of seasonal and interannual variations. I would like to have seen them for lengths of a day, too.

Reply: We added numerical correlation values between the parameters for different heights and different locations, whereby we want to mention that these values corresponds to the whole data set and not for short time series (below one/half year). With our current methods we do not find a connection between the LOD and the zonal MLT wind on time scales less than a year.

We added to the text:

Additionally, we show in Table 1 correlation coefficients for the 4 locations for the altitudes between 80 and 98 km. Positive correlation values correspond to the occurrence of an eastward directed wind together with an increased LOD. The values of the NH follow a similar pattern, with positive coefficients below the vertical transition height and negative above. Davis shows a different pattern, with overall negative correlation coefficients. This relies in the opposite zonal wind pattern compared to the NH. Theoretical, a time shift of $\sim$ half a year would lead to a similar correlation pattern as the NH.

According to Abarca-del Rio et al. (2003) an accurate estimation of the impact of the solar radiation is quite complicated, due to the point that internal oscillations in the climate system show variations within the same frequency as the 11 year solar cycle. Further, Gray et al. (2010) supports this statement and mention that the problem is further caused due to the small influence of the solar forcing on the climate. Nevertheless, Chapanov and Gambis (2008) showed that based on a decomposition of the LOD, the solar activity (10.47 years) is included.

Abarca-del Rio, R., Gambis, D., Salstein, D., Nelson, P., and Dai, A.: Solar activity and earth rotation variability, Journal of geodynamics, 36, 423–443, https://doi.org/doi:10.1016/S0264-3707(03)00060-7, 2003.

Gray, L. J., Beer, J., Geller, M., Haigh, J. D., Lockwood, M., Matthes, K., Cubasch, U., Fleitmann, D., Harrison, G., Hood, L., Luterbacher, J., Meehl, G. A., Shindell, D., van Geel, B., and White, W.: Solar influence on climate, Reviews of Geophysics, 48, https://doi.org/10.1029/2009RG000282, http://dx.doi.org/10.1029/2009RG000282, 2010.

Chapanov, Y. and Gambis, D.: Correlation between the solar activity cycles and the Earth rotation, Proc. Journées 2007 "Systèmes de référence Spatio-Temporels", edited by: Capitaine, N., Obs. de Paris, 206–207, 2008.

Zonal mean winds are presented using meteor radar measurements at 3 sites in the northern hemisphere and the high latitude southern hemisphere, and Aura/MLS. I expected that Aura/MLS data results are compared with radar results for a validation and then would present any global variations. However, Aura/MLS results are used only for comparisons of radar results in the northern hemisphere. Because authors conclude zonal mean winds agree between MLS/Aura and radar measurements, I do not understand motivation of presenting trends of lengths of a day and zonal mean winds from both meteor radars (Figure 6) and Aura/MLS (Figure 7).

Reply: Based on MLS data we added a Figure where is shown the zonal mean winds at ∼80 km geometric height together with the LOD at the fixed longitude 0°-20°E between northern high latitudes and southern high latitudes. With this Figure we show the development of the global variations between both parameters. As result can be seen that the correlation between the LOD and the mean zonal wind increase towards the northern high latitude. The same would be seen if half a time shift of ∼ half a year would be added to the time series.

FIGURE 9

[Figure]

Caption Figure9 : Zonal MLS wind (red) and LOD (black) at ~80km geometric height for 0°-20°E. The left part show the values for the southern hemisphere, the right for the northern hemisphere, for every 10° latitude.

Further we added some text:

Figure 9 shows, based on MLS data, the zonal mean wind at ~80 km geometric height and the LOD. These zonal mean winds include wind values within the longitude grid between 0°E and 20°E, which is comparable to the NH stations. The Figure is divided in 10° latitude steps from 80° to 10°S/N. Each latitude grid includes values for +/- 6°. For the MLS observations the comparison between the wind and the LOD are similar to the 80 km meteor results at the respective latitudes. Further can be seen the occurrence of half a year time shift, between both high hemispheres. A 180° phase shift would lead to the wind-LOD pattern of the opposite hemisphere. Furthermore, the strongest correlation can be seen between both parameters at northern polar latitudes. Due to an increase in the difference between the geometric and geopotential heights, we do not show comparison for higher altitudes. Further we added correlation coefficient between the zonal mean wind and the LOD for each latitude. As result occurs a correlation increase towards the northern high latitude. The same could be seen if a 180° phase shift is added to the time series.

Lastly, authors compare zonal mean winds using Andenes and Davis in the northern and southern hemisphere in a same season and conclude that a difference is caused only by lengths of a day between northern hemisphere summer and winter. However, I believe that mean winds from ground measurement only one site include zonal mean wind and stationary planetary waves and difference of stationary planetary wave amplitudes largely results to a difference of mean winds of ground measurements. I also believe that main reason of interhemispheric differences in atmospheric dynamics is a difference of topography. It makes a difference of atmospheric waves with interact with mean winds. The difference of topography makes an interhemispheric difference of chemical composition in the atmosphere, such as water vapour, ozone, and carbon

dioxide, which makes an interhemispheric difference of viscosity and then winds.

Reply: We totally agree with that point. Our aim is not to state that the differences in the mesospheric zonal mean winds are only a result of the LOD. As we already wrote, we only want to point out, that beside other factors, which have a way stronger influence on the differences of the wind, the LOD has an influence on the winds on both hemispheres. As example for other influencing factors can be called the topography, chemical components and the occurrence and propagation of gravity waves. These waves are the main drivers of the atmospheric wind circulation and therefore also influences the local wind differences at both hemispheres. Unfortunately based only on wind measurements we are not able to estimate a precise value on how strong the connection is between zonal mean wind the LOD.

We added/reformulated some text in the manuscript to clarify the point in the manuscript.

We have to note that beside many others factors, this is only one reason, and by far not the dominant factor, for the wind differences between both locations at theses altitudes. Other physical processes have also a strong effect on the hemispheric wind differences e.g., the topography, chemical composition of the (Marsh (2007), Lee (2018)), and the occurrence and propagation of gravity waves. These waves are the main drivers of the atmospheric wind circulation and therefore also influence the local wind differences at both hemispheres. Furthermore gravity waves lead, compared to the annual mean, to a colder summer mesosphere and a warmer winter mesosphere e.g., Luebken (2014). These temperature differences also fit well to the atmospheric expansion/shrinking. Unfortunately, based only on wind measurements we are not able to estimate a precise value on how strong the connection is between zonal mean wind with the LOD. For a more detailed understanding of these phenomena global density observations would be required.

Lübken, F.-J., Höffner, J., Kaifler, B., and Morris, R., J.: Winter/summer mesopause

temperature transition at Davis (69°S) in 2011/2012, Geophys. Res. Lett., 41, 5233–5238, https://doi.org/10.1002/2014GL060777, 2014.

Marsh, D., R., Garcia, R., R., Kinnison, D., E., Boville, B., A., Sassi, F., Solomon, S., C., and Matthes, K.: Modeling the whole atmosphere response to solar cycle changes in radiative and geomagnetic forcing, Journal of geophysical research, 112, https://doi.org/doi:10.1029/2006JD008306, 2007.

Lee, J., N., Wu, D., L. R. A., and Fontenla, J.: Solar cycle variations in mesopheric carbon monoxide, Journal of atmospheric and solarterrestial physics, 170, 21–34, https://doi.org/https://doi.org/10.1016/j.jastp.2018.02.001, 2018.

I report that this manuscript needs further consideration and discussion.

Specific comments

Line 24 on page 4: What is d? Because equation (7) shows d(t), it must be a variable parameter depending on time (I expect that t stands for time). Can authors change "d" to another symbol or acronym because it is very confusing with integral and differential symbols?

Reply: d which we changed now to D is the angular velocity of the Earth. To avoid misunderstandings we didn't choose $\omega$, because it is already used in the equations 1, 5 and 6 as angular velocity for an altitude defined atmospheric layer.

Lines 21 to 24 on page 5: I am suspicious if you can estimate winds at 78 (or below ∼85 km) and 100 km by meteor radars. What is an altitude resolution, and every how much in km did authors determine hourly mean winds? Is there any threshold for a determination, such as elevation angle, range, minimum and maximum radial velocities, and minimum number of sampling meteor echoes? Although authors mention uncertainties as "between 2 and 6 m/s", weightings of uncertainties are very different between 50 m/s wind with 6 m/s uncertainty and 5 m/s with 6 m/s uncertainty.

Reply: Some literature to show results for the wind estimation based on specular meteor trails:

Hall, C., Aso, T., Tsutsumi, M., Nozawa, S., Meek, C., and Manson, A.: Comparison of meteor and medium frequency radar kilometer scale MLT dynamics at 70_ N, J. Atmos. Sol.-Terr. Phys., 68, 309–316, https://doi.org/10.1016/j.jastp.2005.03.025, 2006.

Hocking, W. K., Fuller, B., and Vandepeer, B.: Realtime determination of meteor-related parameters utilizing modern digital technology, J. Atmos. Sol.-Terr. Phys., 69, 155–169, https://doi.org/10.1016/S1364-6826(00)00138-3, 2001a.

Jacobi, C., Arras, C., Kürschner, D., Singer, W., Hoffmann, P., and Keuer, D.: Comparison of mesopause region meteor radar winds, medium frequency radar winds and low frequency drifts over Germany, Adv. Space Res., 43, 247–252, https://doi.org/10.1016/j.asr.2008.05.009, 2009.

McCormack, J., Hoppel, K., Kuhl, D., de Wit, R., Stober, G., Espy, P., Baker, N., Brown, P., Fritts, D., Jacobi, C., Janches, D., Mitchell, N., Ruston, B., Swadley, S., Viner, K., Whitcomb, T., and Hibbins, R.: Comparison of mesospheric winds from a high-altitude meteorological analysis system and meteor radar observations during the boreal winters of 2009/2010 and 2012/2013, J. Atmos. Sol.-Terr. Phy., https://doi.org/10.1016/j.jastp.2016.12.007, 2016.

Stober, G., Matthias, V., Jacobi, C., Wilhelm, S., Höffner, J., and Chau, J. L.: Exceptionally strong summer-like zonal wind reversal in the upper mesosphere during winter 2015/16, Ann. Geophys., 35, 711–720, https://doi.org/10.5194/angeo-35-711-2017, 2017.

Wilhelm, S., Stober, G., and Chau, J. L.: A comparison of 11-year mesospheric and lower thermospheric winds determined by meteor and MF radar at 69_ N, Annales Geophysicae, 35, 893–906, https://doi.org/10.5194/angeo-35-893-2017, https://www.ann-geophys.net/35/893/2017/, 2017.

The monostatic meteor radars cover an altitude range between 75 and 110 km and

the obtained winds have an hourly temporal resolution and a vertical altitude resolution of 2 km in the applied analysis. Within these altitudes, we are able to detect meteors whereby qualitative good wind measurements are reached between 78 and 100 km. Below 75 km we are limited due atmospheric conditions and above 110 km due to technical limitations.

In these time and height window each meteor is weighted by its statistical uncertainty and by its temporal distance from the centre of the window by using a Gaussian kernel. Further regularization is implemented in the wind estimation, which allows estimating the wind within the windows by having at least 3 meteors. As example in December 2015 we detected ∼90.000 meteors between 78 and 100 km for the location of Andenes. These meteors follow a Gaussian height distribution, which leads to detections of ∼300 meteors at 90+/-1 km altitude window per hour. At 90 km these meteors are detected within an observational diameter of 425 km and all detected meteors within the diameter are taken for the wind analysis. Of course there are thresholds for the determination, as e.g. elevation angle of zenith < 65°. Further details can be found in then mentioned literature.

Depending on the amount of available detected meteors within the window, the statistical uncertainties of the meteor wind measurements vary between 2 and 6 m/s, whereby values larger than 4 m/s nearly only be reached at the edges of the observation range. In Wilhelm et al. (2017) is shown in Figure 1 an altitude/time distribution of the uncertainties. There its shown that based on the meteor altitude distribution, which includes daily as well as seasonal variations, the statistical uncertainties vary between 2 and 4 m/s between 84 and 94 km.

We don't want to describe the complete wind analysis within this manuscript, therefore we linked to Stober et al. (2017) as well as to Hocking et al. (1999) where the analysis are described in detail.

Line 31 on page 5: Please check a vertical resolution. In my knowledge, Aura/MLS

data are every 1.3 km up to 50 km, 2.7 km up to 62 km and 5.4 km above.

Reply: The vertical resolution which we use in this study is ∼4 km in the stratosphere and ∼14 km in the mesosphere. We added a reference.

Livesey, N.J., Read, W.G., Lambert, A., Cofield, R.E., Cuddy, D.T., Froidevaux, L., Fuller, R.A., Jarnot, R.F., Jiang, J.H., Jiang, Y.B., Knosp, B.W., Kovalenko, L.J., Pickett, H.M., Pumphrey, H.C., Santee, M.L. , Schwartz, M.J., Stek, P.C., Wagner, P.A., Waters, J.W., Wu, D.L., 2007. EOS MLS Version 2.2 Level 2 Data Quality and Description Document. Technical Report, Version 2.2 D-33509, Jet Propulsion Lab., California Institute of Technology, Pasadena, California 91198-8099.

Matthias, V., Hoffmann, P., Rapp, M., and Baumgarten, G.: Composite analysis of the temporal development of waves in the polar {MLT} region during stratospheric warmings, J. Atmos. Sol.-Terr. Phy., 90–91, 86–96, https://doi.org/10.1016/j.jastp.2012.04.004, 2012.

Line 18 on page 6: Juliusruh and Collm are at nearly same location in a global sense. What causes a difference of reversal altitudes by 3 km? Are they systematic difference?

Reply: Even if, on a global sense, Collm and Juliusruh are nearly located on the same latitude, small changes at mid and especially lower latitudes can show strong differences in the transition height. Even if it is not included in this paper, we further compared the mid latitude data with meteor radar data from a Canadian location (43.3 °N), with the result of an even deeper (80 km and blow) vertical wind shear.

Lines 9 to 10 on page 8: How was "the fluctuation in the LOD" obtained? Was it by equation (7)? If so, what is d(t), as asked above? Was d(t) obtained from measurements or some simulation models?

Reply: The LOD data are the result from a combination of several intra-technique services, each associated with a given space geodetic technique. One of them is the VLBI technique, which is able to determine the celestial pole and the Earth rotation

angle and therefore observes changes in the day lengths. Measurements derived by VLBI consist of simultaneous observations of extra-galactic radio sources by two or more radio telescopes. During a standard VLBI observation of 24 hours, three to eight globally distributed telescopes observe up to 60 extra-galactic radio sources. These sources are located in a distance of 2-12 billion light-years and emit broadband microwave signals which can be assumed as a plane wave front when they arrive at the Earth. These radio sources are e.g., quasars which are active galactic nucliis of very high brightness, and which are so far away that no proper motion of them has ever been observed. Therefore they serve as best available fixed position to a fixed reference. Any change in the Earth's spinning or in the Earth orientation, measured by extra-galactic signals can be determined within a fraction of a millisecond of arc. The use of interferometry between several stations leads to the fundamental geodetic VLBI information. Therefore the LOD can be defined by equation (7).

Thomas, J.: An Analysis of Long Baseline Interferometry. DSN Progress Report, JPL Techniqcal Report, 8, 32–1526, 1972.

Campbell, J.: Very long baseline interferometry., pp. 67–87, Berlin Springer Verlag, ttps://doi.org/doi:10.1007/BFb0010105, 1987.

Boeckmann, S.: Robust determination of station positions and Earth orientation parameters by VLBI intra-technique combination, Ph.D. thesis, Friedrich-Wilhelms-University, http://hss.ulb.uni-bonn.de/diss_online, 2010.

Line 22 on page 8: What is "the F10.7 solar cycle"? Is it the 11-year cycle, the 27-day cycle, or both cycles?

Reply: The F10.7 solar cycle is the 11 –year solar cycle. We further tried to estimate the 27-day-cycle within the zonal winds, but didn't found any correlation. We changed in the manuscript F10.7 solar cycle to F10.7 11-year solar cycle.

Line 29 on page 8: LOD (either length of a day or fluctuation in a length of a day)

must have unit of time (probably second from Figures 6 to 8). Why is an LOD unit ms (millisecond or meter times second)?

Reply: ms = milliseconds, we added this in the text.

Line 33 on page 8: Again, please make sure what "the solar cycle" is, 11 year, 27 day, both, or some other cycle? Also, how much does "the solar cycle" influence on a fluctuation of a length of a day? It means how much important to remove a solar cycle influence.

Reply: We added some text to clarify this part: (see 1st point) According to Abarca-del Rio et al. (2003) an accurate estimation of the impact of the solar radiation is quite complicated, due to the point that internal oscillations in the climate system show variations within the same frequency as the 11 year solar cycle. Further, Gray et al. (2010) supports this statement and mention that the problem is further caused due to the small influence of the solar forcing on the climate. Nevertheless, Chapanov and Gambis (2008) showed that based on a decomposition of the LOD, the solar activity (10.47 years) is included.

Abarca del Rio, R., Gambis, D., Salstein, D., Nelson, P., and Dai, A.: Solar activity and earth rotation variability, J. Geodyn., 36, 423–443, 2003.

Gray, L. J., Beer, J., Geller, M., Haigh, J. D., Lockwood, M.,Matthes, K., Cubasch, U., Fleitmann, D., Harrison, G., Hood, L., Luterbacher, J., Meehl, G. A., Shindell, D., van Geel, B., and White, W.: Solar influences on climate, Rev. Geophys., 48, RG4001, doi:10.1029/2009RG000282, 2010.

Chapanov, Y. and Gambis, D.: Correlation between the solar activity cycles and the Earth rotation, Proc. Journées 2007 "Systèmes de Référence Spatio-Temporels", edited by: Capitaine, N., Obs. De Paris, 206–207, 2008. (https://syrte.obspm.fr/jsr/journees2007/pdf/s4_18_Chapanov.pdf)

Line 12 on page 9: What is "the size range"?

Reply: we corrected the sentence.

Line 25 on page 9: I do not see that the authors showed effects on mesospheric winds by expansion/shrinking of the upper atmosphere in this work.

I saw that the authors showed correlations between zonal winds in the mesosphere and fluctuations in a length of a day. Stober et al. inferred that fluctuations in a length of a day are correlated with a variation of a thickness of the upper atmosphere. It is possible to expect that zonal winds in the mesosphere are related to a thickness of the upper atmosphere. Please revise it.

Reply: We added Figure 4 to show a direct connection between fluctuation in the solar radiation and the zonal wind. Based on this and on the results of Emmert et al. (2004) and Stober et al. (2012) we show the relation between the thickness of the upper atmosphere and the prevailing zonal wind.

We added the text:

To underline this statement, Figure 4 shows, for the location of Andenes, the zonal mean wind between 84 and 94 km together with the F10.7 11-year solar cycle index (black line). An enhancement of the eastward directed wind occurs together with a stronger F10.7 index and more clearly an increase of the westward directed wind together with a smaller F10.7. Furthermore a shift occurs in the summer vertical wind shear, which is also correlated with the solar cycle, whereby a shift to higher altitudes takes place together with a decrease of the solar radiation, due to a change in the neutral density.

FIGURE4

Caption Figure4: Zonal mean wind for Andenes for the heights between 84 and 94 km, together with the F10.7 11 year solar cycle index in black.

Emmert, J., T., Picone, J., M., Lean, J., L., and Knowles, S., H.: Global change in the thermosphere: Compelling evidence of a secular decrease in density, Journal of

geophysical research, 109, doi:doi:10.1029/2003JA010176, 2004.

Stober, G., Jacobi, C., Matthias, V., Hoffmann, P., and Gerding, M.: Neutral air density variations during strong planetary wave activity in the mesopause region derived from meteor radar observations, Journal of Atmospheric and Solar-Terrestrial Physics, 74, 55–63, doi:10.1016/j.jastp.2011.10.007, 2012.

Figure 2 and 3: Why do they exclude Davis in Figure 2 and southern hemisphere? Reply: We will add Davis, as well as the southern hemisphere (MLS) in Figure 2 / 3. Further we will add for both locations a description in the text.

Figure 2 and 3 captions: Correct to "positive". Reply: corrected

Figure 6: Please describe what black and blue dashed lines are in a caption. Reply: description is added in the caption.

Technical corrections

General Reply: We thank the referee for the advices. We will correct the mentioned points, and added here for some points few comments for the Referee.

Line 10 on page 1: Do authors use LOD as "length of a day" or "fluctuations in a length of a day" in this manuscript? Line 24 on page 4 says "length of day (LOD)". However, "LOD" is used in a subsection 3.2 and a caption of Figure 6 although most of them must imply "fluctuation of LOD", except for lines from 9 to 10 on page 8 say "fluctuation in the LOD". Please make it consistent.

Reply: LOD is the acronym for the "fluctuation in a length of a day", which is also used as this in the area of geodesy. We added and corrected parts to make this more clear.

Lines 9 to 10 on page 2: "shrinking of the middle atmosphere between solar minimum and solar maximum" is very confusing. Does the middle atmosphere shrink at the solar minimum, the solar maximum, or both at the solar minimum and maximum?

Reply: We reformulated the sentence and added some more references:

Previous studies as, e.g., Walterscheid (1989), Marsh et al (2007), Emmert (2015), and Lee et al. (2018) showed that solar cycle variations affects the atmospheric density, temperature, chemical composition and winds over the whole atmosphere, but in particular, in the MTI (Mesosphere-Thermosphere-Ionosphere) system.

Later on we wrote that Emmert et al. (2010) showed compared to an average over some solar cycles a decrease in the neutral density during a solar minimum.

Walterscheid, R., L.: Solar Cycle effects on the upper atmosphere: Implications for Satellite Drag, Journal of spacecraft and rockets, 26, 439–444, https://doi.org/DOI: 10.2514/3.26089, 1989.

Marsh, D., R., Garcia, R., R., Kinnison, D., E., Boville, B., A., Sassi, F., Solomon, S., C., and Matthes, K.: Modeling the whole atmosphere response to solar cycle changes in radiative and geomagnetic forcing, Journal of geophysical research, 112, https://doi.org/doi:10.1029/2006JD008306, 2007.

Emmert, J. T.: Altitude and solar activity dependence of 1967-2005 thermospheric density trends derived from orbital drag, Journal of geophysical research: space physics, 120, 2940–2950, https://doi.org/doi:10.1002/2015JA021047., 2015.

Lee, J., N., Wu, D., L. R. A., and Fontenla, J.: Solar cycle variations in mesopheric carbon monoxide, Journal of atmospheric and solar terrestial physics, 170, 21–34, https://doi.org/https://doi.org/10.1016/j.jastp.2018.02.001, 2018.

Line 19 on page 2: Does "conversation" mean "conservation"? yes, it does. -corrected-

Line 8 on page 5: What does "on an in average" mean? – we modified the sentence -

Line 4 on page 6: I feel that "combined 04 data from the international Earth Rotation and Reference System Service (IERS)" is more appropriate. –corrected-, Thanks for the advice.

Line 24 on page 6: "qualitatively"? – corrected -

Lines 24 to 25 on page 6: I do not understand the sentence and suggest revision. – Reply: we deleted the sentence, because it causes confusion and were not needed.

Line 30 on page 6: Are MLS data shown in a geopotential height? If so, why "above 90 km" is suddenly described in geometric height?

Reply: The geometric altitudes are approximately estimated from the pressure levels as described in Matthias et al. (2013): h = -7 * ln(p/1000), where h is the altitude in km and p the pressure in hPa. We are aware that there is difference between the geometric and the geopotential heights especially in the MLT. Furthermore, we neglect altitudes above 85 km geometric height for closer investigations, because the obtained winds show larger discrepancies to the local radar measurements. We only used the upper heights for a general validation based on composites to show similarities.

Matthias, V., Hoffmann, P., Manson, A., Meek, C., Stober, G., Brown, P., and Rapp, M.: The impact of planetary waves on the latitudinal displacement of sudden stratospheric warmings, Ann. Geophys., 31, 1397-1415, https://doi.org/10.5194/angeo-31-1397-2013, 2013.

Added text: The geometric heights are approximately estimated from pressure levels as described in Matthias (2013): h = -7* ln(9/1000), where h is the altitude in km and p the pressure in hPa. Furthermore, we are aware about a difference between the geometric and geopotential heights, which increase especially above 80 km. Therefore, we focus in this work on the height range between 60 and 80 km . . .

Lines 6 to 7 on page 7: It is very ambiguous. Does a density increase occur in summer OR winter, and at the solar minimum OR maximum?

Reply: Figure 4 is a theoretical approach to show changes in the rotation speed, for a defined atmospheric layer, based on changes in the density. For this approach it doesn't matter when the density increase/decrease occurs, it only show the results based on the theoretical change. Nevertheless, according to Emmert (2010) occurs a

density decrease during the time of a solar minimum and a density increase during a solar maximum, respectively.

We reformulated the text: The density increase takes place for longer time scales during a solar maximum (e.g., Emmert et all, 2010) and on annual time scales during the winter, when the Earth-Sun distance is smaller. Both cases influence the temperature within this atmospheric layer as well as their expansion compared to the annual mean. Overall the density variation during an 11-year solar cycle are stronger than the variation causes due to Earth-Sun distance.

Emmert, J. T., Lean, J. L., and Picone, J. M.: Record-low thermospheric density during the 2008 solar minimum, Geophysical Research Letters, 37, n/a–n/a, https://doi.org/10.1029/2010GL043671, http://dx.doi.org/10.1029/2010GL043671, l12102, 2010.

Line 16 on page 7: Change "the northern and the southern hemisphere" to "the northern and southern hemispheres". –corrected-

Line 16 on page 7: Remove comma between "opposite" and "fluctuations". –corrected-

Lines 20 to 21 on page 7: What is "between two locations on the same latitude"? Does it mean "at the same latitude in the northern and southern hemispheres"?

Reply: we reformulated the sentence: Therefore the prevailing wind within the MLT region should be similar in magnitude between Andenes and Davis, which are located at the same latitude in the northern and southern hemispheres.

Line 21 on page 8: It should be "additionally". –corrected-

Line 22 on page 8: It should be "relatively". –corrected-

Line 29 on page 8: Please make sure if "seasonal fluctuation" means "seasonal variation of a fluctuation", "seasonal means of fluctuation", or something else. –corrected, Thanks for the advice-

Line 33 on page 8: What does "as result as" mean? –corrected-

Line 19 on page 9: "This reversal can be explain can be explain" must be "This reversal can be explained". –corrected-

Lines 20 to 21 on page 9: First, "station" on line 21 must be "stations"? What are "the polar and the second midlatitude stations"? "the polar stations" include both Andenes and Davis? Is "the second milatitude station (I think not "stations" in this case)" Juliusruh or Collm? Did the authors define "first" and "second" stations previously?

Reply: We reformulated the part according comments of Ref #1: In the Figures 10 and 11 are shown long term changes of annual LOD (black) and annual zonal mean winds (red) for Collm and for Davis. At this point, we have to mention that a tendency over a long time series is not linear in time. Parameter which influence the tendency of the wind and the LOD also vary over time and therefore be observed in long time series should be limited within a specific period. Such changes are often be approximated by a piecewise linear trend model (e.g., Tomé and Miranda (2004), Merzlyakov et al. (2009) and Jacobi et al. (2011)), where different linear fit tendencies are estimated for different time periods. Nevertheless, due to the length of the available data series we decide to use no piecewise linear trend model. The wind values exclude seasonal and solar cycle variations and the LOD excludes the seasonal variations. Exemplary for the locations of Collm (Figure 10) the altitudes between 80 and 96 km are displayed. The errorbars corresponds to the annual variance for each height and the dotted lines show the long term tendency for each parameter. The result is that a long term increase of the LOD occurs together with a long term decrease of the zonal wind. Above 94 km the tendency reverses for the mid latitude locations into a slightly positive wind. This reversal can be explain by the stronger influence due to gravity wave filtering, which has to be considered and cannot be excluded by filtering the data. The tendencies of an increased value for the LOD and a decreased value for the zonal mean wind can be seen for all mid latitude locations and also for Davis (see Figure 11). Andenes shows for all altitudes increase tendency in the zonal wind. The results indicates that

the connection between the LOD and the wind are more pronounced at lower latitudes, which simply explainable by the rotation velocity, which is higher at the middle latitude stations than at the polar latitudes like Andenes and Davis. The results of an increase of the LOD and a decrease of zonal wind fits to the relation between fluctuations in the neutral density and the zonal wind, as shown Stober et al. (2012).

Tomé, A., R. and Miranda, P., M. A.: Piecewise linear fitting and trend changing points of climate parameters, Geophys. Res. Lett., 31, https://doi.org/doi:10.1029/2003GL019100, 2004.

Merzlyakov, E., G., Jacobi, C., Portnyagin, Yu., I., and Solovjova, T., V.: Structural changes in trend parameters of the MLT winds based on wind measurements at Ob-ninsk (55°N, 37°E) and Collm (52°N, 15°E), Journal of atmospheric and solar-terrestial physics, 71, 1547–1557, https://doi.org/doi:10.1016/j.jastp.2009.05.013, 2009.

Jacobi, C., Hoffmann, P., Liu, R., Q., Merzlyakov, E., G., Portnyagin, Yu., I., Manson, A., H., and Meek, C., E.: Long-term trends, their changes, and interannual variability of Northern Hemisphere midlatitude MLT winds, Journal of Atmospheric and Solar-Terrestrial Physics, 75-76, 81–91, https://doi.org/doi:10.1016/j.jastp.2011.03.016, 2011.

Line 1 on page 10: "hemisphere" must be "hemispheres". –corrected-

Lines 6 to 7 on page 10: Why do authors specify "the middle latitude stations" as Collm and Juliusruh? Is "the polar station" only Andenes? How about Davis? Reply: We added to location of Davis, and corrected the sentence if needed

Line 10 on page 10: "not figured out" must be "not be figured out". –corrected-

Line 13 to 14: I do not understand this sentence. Please revise it. Reply: For a better understanding we partly reformulated the conclusion.

Line 1 on page 11: "ssignal" must be "signal". –corrected-

Line 20 on page 11: "datadata" must be "data". –corrected-

Please also note the supplement to this comment:
https://www.ann-geophys-discuss.net/angeo-2018-51/angeo-2018-51-AC2-supplement.zip

[Figure]

[Figure]

**Fig. 1.**

Fig. 2.

---

## Referee Report (RR1)

Review of revised angeo-2018-51, "Connection between the length of day and wind measurements in the mesosphere and lower thermosphere and mid and high latitudes"

**General comments**

The manuscript is revised very well. My main concern in the original one was it might impress that the authors concluded that fluctuations in the length of a day (LOD) caused interhemispheric variations of mean zonal winds. Because the topic is not conclusive, rather introducing a new insight, it needs to be very careful that the LOD may be one of possible sources of interhemispheric variations.

This interesting topic in the manuscript addresses further curious questions to readers, correlations of the LOD with mean meridional winds and tidal amplitudes while it may be very difficult to analyze correlations with tidal phases and periods due to very small values if they exist. I expect that the authors have already had them for future work.

Adding correlations of the LOD with global mean zonal winds from MLS satellite observations presents very interesting, why the correlations are not consistent globally and it is probably because the LOD is not the only source for interannual variations. My only one concern is that mean winds from a radar at one site is a superposition of zonal mean and stationary planetary wave winds. Using satellite data at nearby longitudes is fine for comparisons with radars. But I would like authors to present global correlations by averaging data at all longitudes, so that correlations with zonal mean zonal winds are seen by cancelling stationary planetary waves.

**Specific comments**

Page 1, Line 8: Does a value "~4 m/s" a critical value regardless locations (latitude and latitude) and time (season)? If not, please specify conditions for this value.

Page 1, Line 14: "show" must be "shows".

Page 1, Line 17: "zonal mean wind" must be "mean zonal wind".

Page 2, Line 3: "inverse" must be "inversely".

Page 2, Line 10: "affects" must be "affect".

Page 2, Line 12: Add "by" between "simulation" and "Marsh".

Page 2, Line20: "show" must be "showed".

Page 3, Line 3: Add "to" between "according" and "Trenberth".

Page 5, Line 1: Replace "of" by "around".

Page 5, Line 2: "fluctuations in the day length" can be replaced by "LOD".

Page 5, Line 29: "hemisphere" must be "hemispheres".

Page 6, Line 27: "station" must be "stations".

Page 6, Line 28: Add space between "Davis" and "(".

Page 8, Line 7: "influences" must be "influence".

Page 8, Line 8: "are" must be "is".

Page 8, Line 9: "causes" must be "caused".

Page 8, Line 32: What does "this is only one reason"?

Page 10, Line 20: Remove "the" between "quite" and "opposite"?

Page 11, Lines 2, 5 and 16: "zonal mean wind" must be "mean zonal wind".

Page 11, Line 7: Remove "be".

Page 11, Line 11: "locations" must be "locations".

Page 11, Line 14: "explain" must be "explained".

Page 11, Line 18: "are" must be "is"

Page 11, Line 20: "fits" must be "fit".

Page 11, Line 25: Does "smaller" mean "shorter"?

Page 12, Line 9: "theses" must be "these".

Page 12, Line 13: "Additional" must be "Additionally".

Page 12, Line 17: Remove "way".

Page 12, Line 18: "point on" is "point out"?

---

## Referee Report (RR2)

**Re-Review of:** "Connection between the length of day and wind measurements in the meaosphere and lower thermosphere at mid and high latitudes.
by Sven Wilhelm et al. [AnGeo 2018-15 rev., rcvd Oct. 2018]

My previous comments on the original version criticisms have well answered - but there have been interesting new items added. So I have a few more comments.

According to the theory here, the expansion and contraction of the atmosphere based on the distance from the sun should influence the seasomal changes in the LOD. So the LOD should lag the heating - that is heating causes expansion, the atmosphere slows down (maintaining conservation of angular momentum) and since the earth is turning eastward, that means there will be a westward perturbation in the atmospheric wind. The correlations presented in Table 1. appear to present an ideal way to test this lag. Is there a lag, and if so, is it in the expected sense? There are 10 years of data available; if the theory is correct, some effect of lag on correlation should be seen.

Pg 5 lines 3-8 and equation 7: This is not clear. I take "astronomically determined" to mean D is "siderial" angular velocity, which results in ∼4 min. per day rotation time (∼86164 sec.) less than "mean solar day" (defined as 86400 sec.). In this case the "LOD" as defined by equation 7 will always be negative.
Alternately, if D refers to "solar day" - then there is another daily/seasonal non-rotational factor related to the changing speed of the Earth in its slightly elliptical orbit.

Pg. 7 line 32-34: The statement says an enhanced eastward directed wind is linked to an increased F10.7 index. But presumably increased F10.7 means an expanded atmosophere which means slower atmospheric rotation, i.e. reduced eastward?

Much argument is expended in this paper to show that seasonal changes in LOD and zonal wind are expected due to the effect of changes in Sun-Earth distance on the atmosphere. In the abstract it should be stated clearly, as it is in the conclusion (Pg. 12, line 10,11) that these seasonal changes were not found in the wind and LOD data probably because of competing effects, such as ... (that is, unless lagged correlation show anything interesting.)
Minor typos etc.:

Pg. 8 line 5: "... is higher than during ... "

Pg. 9 line 20: "explicitly"

Pg. 11 line 20: ."... zonal wind agrees with the relation ..." ?

Pg. 12 line 9: ".... between these ..."

Pg. 12 Line 14: "Additionally, ... "
Pg. 12 line 16: "Further we only compare ..."

Pg. 12 line 17: "... effects which drive ..."

$$-///-$$

---

## Author Response (AR2)

Review #1

Review of revised angeo-2018-51, "Connection between the length of day and wind measurements in the mesosphere and lower thermosphere and mid and high latitudes"

**General comments**

The manuscript is revised very well. My main concern in the original one was it might impress that the authors concluded that fluctuations in the length of a day (LOD) caused interhemispheric variations of mean zonal winds. Because the topic is not conclusive, rather introducing a new insight, it needs to be very careful that the LOD may be one of possible sources of interhemispheric variations.

This interesting topic in the manuscript addresses further curious questions to readers, correlations of the LOD with mean meridional winds and tidal amplitudes while it may be very difficult to analyze correlations with tidal phases and periods due to very small values if they exist. I expect that the authors have already had them for future work.

**General reply:**
We appreciate the work which was done by the referee to increase the quality of the manuscript.

1. Adding correlations of the LOD with global mean zonal winds from MLS satellite observations presents very interesting, why the correlations are not consistent globally and it is probably because the LOD is not the only source for interannual variations. My only one concern is that mean winds from a radar at one site is a superposition of zonal mean and stationary planetary wave winds. Using satellite data at nearby longitudes is fine for comparisons with radars. But we would like authors to present global correlations by averaging data at all longitudes, so that correlations with zonal mean zonal winds are seen by cancelling stationary planetary waves.

**Reply:** We added global correlations by averaging zonal mean wind data over all longitude in the Figure (green numbers), as well as, in some comments the text. The shape of the curves between the global average and the previous average between 0-20°E are nearly equal, therefore we didn't add them in the Figure.

Added /modified text : We added correlation coefficients (black) between the mean zonal wind and the LOD for each latitude. A correlation increase towards the northern high latitudes is visible. The same would be seen if a 180° phase shift is added to the time series. Additionally, we present global correlations (green) by averaging mean zonal wind data over all longitudes, whereby possible stationary planetary waves are filtered. The global correlation coefficients are nearly similar to the values for previous average winds between 0-20°E. The shape of the curves between the global average winds are also are nearly equal, therefore we didn't add them in the Figure.

[Figure]

**Caption:** Zonal MLS wind (red) and LOD (black) at ~80 km geometric height for 0°-20°E. The left part show the values for the southern hemisphere, the right for the northern hemisphere, for every 10° latitude. The black correlation coefficients (r) are estimated for the mean between 0°-20°E, and the green coefficients corresponds to global average over all longitudes.

**Specific comments**

Page 1, Line 8: Does a value "~4 m/s" a critical value regardless locations (latitude and latitude) and time (season)? If not, please specify conditions for this value.

**Reply:** The above mentioned value was estimated for latitude of 45°. Different latitudes lead to slightly different values for gravity (g). Furthermore, the theoretical estimation of the rotation speed is independent of the longitude and time (season).

We added in section 3.1. additional notes: The calculation is done in 2 km height layers and for the latitude of 45°. Different latitudes lead to slightly different values of g, which is used in equation 4.

We appreciated and corrected the hints regarding the following typos.

Page 1, Line 14: "show" must be "shows".
Page 1, Line 17: "zonal mean wind" must be "mean zonal wind".
Page 2, Line 3: "inverse" must be "inversely".
Page 2, Line 10: "affects" must be "affect".
Page 2, Line 12: Add "by" between "simulation" and "Marsh".
Page 2, Line20: "show" must be "showed".
Page 3, Line 3: Add "to" between "according" and "Trenberth".

Page 5, Line 1: Replace "of" by "around".
Page 5, Line 2: "fluctuations in the day length" can be replaced by "LOD".
Page 5, Line 29: "hemisphere" must be "hemispheres".
Page 6, Line 27: "station" must be "stations".
Page 6, Line 28: Add space between "Davis" and "(".
Page 8, Line 7: "influences" must be "influence".
Page 8, Line 8: "are" must be "is".
Page 8, Line 9: "causes" must be "caused".
Page 8, Line 32: What does "this is only one reason"?
Page 10, Line 20: Remove "the" between "quite" and "opposite"?
Page 11, Lines 2, 5 and 16: "zonal mean wind" must be "mean zonal wind".
Page 11, Line 7: Remove "be".
Page 11, Line 11: "locations" must be "locations".
Page 11, Line 14: "explain" must be "explained".
Page 11, Line 18: "are" must be "is"
Page 11, Line 20: "fits" must be "fit".
Page 11, Line 25: Does "smaller" mean "shorter"?
Page 12, Line 9: "theses" must be "these".
Page 12, Line 13: "Additional" must be "Additionally".
Page 12, Line 17: Remove "way".
Page 12, Line 18: "point on" is "point out"?

Review #2

Re-Review of: "Connection between the length of day and wind measurements in the mesosphere and lower thermosphere at mid and high latitudes."
by Sven Wilhelm et al. [AnGeo 2018-15 rev., rcvd Oct. 2018]

My previous comments on the original version criticisms have well answered - but there have been interesting new items added. So I have a few more comments.

**General reply:**
We appreciate the work which was done by the referee to increase the quality of the manuscript.

1. According to the theory here, the expansion and contraction of the atmosphere based on the distance from the sun should influence the seasonal changes in the LOD. So the LOD should lag the heating - that is heating causes expansion, the atmosphere slows down (maintaining conservation of angular momentum) and since the earth is turning eastward, that means there will be a westward perturbation in the atmospheric wind. The correlations presented in Table 1 appear to present an ideal way to test this lag. Is there a lag, and if so, is it in the expected sense? There are 10 years of data available; if the theory is correct, some effect of lag on correlation should be seen.

**Reply:**
The fluctuations within a year are too weak to be seen in the wind measurements and we are also not able to separate them from the general wind pattern. The influence of the atmospheric waves and large scale geophysical events dominate the wind regime. Furthermore, the solar maximum during the last $24^{th}$ solar cycle was quite weak compared to the previous ones. Under the assumption of a stronger next solar cycle it could be possible to see differences and also a time lag in the wind and also in the LOD during the solar minimum and the solar maximum.

2. Pg 5 lines 3-8 and equation 7: This is not clear. I take "astronomically determined" to mean D is "sidereal" angular velocity, which results in ~4 min. per day rotation time (_86164 sec.) less than \mean solar day" (defined as 86400 sec.). In this case the "LOD" as defined by equation 7 will always be negative.
Alternately, if D refers to \solar day" - then there is another daily/seasonal non-rotational factor related to the changing speed of the Earth in its slightly elliptical orbit.

**Reply:**
For the paper we added the following text:
Within the estimation of the LOD the sidereal time gets converted into solar time, by taking into account the Earth's position, nutation, precession and motion with respect to the stars. Detailed information about the transformation from sidereal time into solar time can be found in e.g., Aoki (1981) and Schnell (2006).

For the referee:

Within this study we don't want explicit explain the transformation from sidereal time into solar time, because it could cause additional questions. Nevertheless, we added additional information, which are mostly cited according the work of Schnell (2006).

Within the estimation of the LOD the sidereal time gets converted into solar time. To explain this we need to go back a step and first need to define the notation of the earth orientation parameters (according IAU1980). The earth orientation parameters (EOP) describe the rotary position of the solid earth relative to a space fixed non-rotation reference system. Here, the EOP is the sum of several consecutive rotations transforming the celestial into the terrestrial system by using according Schnell (2006) and the included references:

$x_{TRS} = W(t) * S(t) * N(t) * P(t) * x_{CRS}$

with $W(t)$ = transformation due to polar motion, $S(t)$ = diurnal earth rotation, $N(t)$ = nutation of the earth, $P(t)$ = Precession. TRS and CRS correspond to the terrestrial and celestial reference system, respectively. In detail $S(t)$ is expressed by

$S(t) = R(GAST)$

With R for a 3-dimentional rotation matrix, and the argument GAST as acronym for the Greenwich Apparent Sidereal Time. This argument corresponds to the current hour angle of the Greenwich meridian relative to the direction of the true spring equinox. The spring equinox varies in time relative to the fixed space coordinates, which is based on $P(t)$ and $N(t)$ and the difference between the true and the mean spring equinox is referred as Equation of Equinoxes (EqE), and the hour angle with respect to the mean equinox is called Greenwich Mean Sidereal Time (GMST). The relation between the mean sidereal time then is

$GAST = GMST + EqE.$

The EqE depends on the current nutation in longitude dp and on the mean obliquity e0:

$EqE = dp * \cos(e0)$

While GMST is referred to the mean spring equinox, the epoch of the solar time UT1 is given by the hour angle relative to the direction of the mean sun. The solar time therefore depends on the diurnal rotation of the earth and additionally on the revolution of the earth around the sun. The true sun is hereby replaced by a mean sun because the motion of the earth is not uniform as a result of the 2[nd] Kepler's Law. The difference between the true and the mean sun is referred as Equation of Time (EqT).

The Greenwich Mean Sidereal Time can be expressed using the solar Universal time as :

$GMST = a_{mSu} - 12h + UT1$

with $a_{mSu}$ as the right ascension of the mean sun. The 12h subtraction are because 0 hr UT1 is defined to be midnight.

From GAST, the solar Universal Time UT1 can be calculated as:

UT1    = GMST – $a_{mSu}$ + 12h

       = GAST – dp * cos(e0) – $a_{mSu}$ +12h

Using the ratio C between the length of a sidereal and a solar time interval:

C = d(GMST) / d UT1 = ( GMST – GMST (UT1 = 0hr) ) / UT1

the relationship between GMST and UT1 can be expressed by:

UT1 = (1 / C) * (GAST – EqE – GMST(0hr UT1)).

On this way the sidereal duration of a day, which is ~1/365 shorter than a mean solar day, gets converted by taking into account the Earth's positon, nutation and precession into the duration of a solar day.

Schnell, D.: Quality aspects of short duration VLBI observations for UT1 determinations, Ph.D. thesis, Rheinische Friedrich-Wilhelms- Universität zu Bonn, http://hss.ulb.uni-bonn.de/2006/0918/0918.htm, 2006.

3. Pg. 7 line 32-34: The statement says an enhanced eastward directed wind is linked to an increased F10.7 index. But presumably increased F10.7 means an expanded atmosphere which means slower atmospheric rotation, i.e. reduced eastward?

**Reply:** That is correct. We appreciate the comment.
The increase of the F10.7 leads to an expanded atmosphere, which further results in a slower atmospheric rotation, and so in e.g., a reduced eastward wind. We removed Figure 4 and the corresponding text, because it was misleading. A changing F10.7 does not have such a strong influence on the density/wind in the shown heights.

4. Much argument is expended in this paper to show that seasonal changes in LOD and zonal wind are expected due to the effect of changes in Sun-Earth distance on the atmosphere. In the abstract it should be stated clearly, as it is in the conclusion (Pg. 12, line 10,11) that these seasonal changes were not found in the wind and LOD data probably because of competing effects, such as ... (that is, unless lagged correlation show anything interesting.)

**Reply:** We added the following text to the abstract:

A direct correlation between the local measured winds and the LOD on shorter time scales cannot clearly be identified, due to stronger influences of other natural oscillations on the wind.

Minor typos etc.:

We appreciated and corrected the hints regarding the following typos.

Pg. 8    line 5: "... is higher than during ... "
Pg. 9    line 20: "explicitly"
Pg. 11 line 20: ."... zonal wind agrees with the relation ..." ?
Pg. 12 line 9: ".... between these ..."
Pg. 12 Line 14: "Additionally, ... "
Pg. 12 line 16: "Further we only compare ..."
Pg. 12 line 17: "... effects which drive ..."

[revised manuscript text omitted]

   Jacobi, C., Arras, C., Kürschner, D., Singer, W., Hoffmann, P., and Keuer, D.: Comparison of mesopause region meteor radar winds, medium frequency radar winds and low frequency drifts over Germany, Advances in Space Research, 43, 247–252,

35    https://doi.org/doi:10.1016/j.asr.2008.05.009, http://dx.doi.org/10.1016/j.asr.2008.05.009, 2009.

   Jones, J., Webster, A. R., and Hocking, W. K.: An improved interferometer design for use with meteor radars, Radio Science, 33, 55–65, https://doi.org/10.1029/97RS03050, http://dx.doi.org/10.1029/97RS03050, 1998.

   Manson, A. H., Meek, C. E., Hall, C. M., Nozawa, S., Mitchell, N. J., Pancheva, D., Singer, W., and Hoffmann, P.: Mesopause dynamics from the scandinavian triangle of radars within the PSMOS-DATAR Project, Annales Geophysicae, 22, 367–386, https://doi.org/10.5194/angeo-22-367-2004, http://www.ann-geophys.net/22/367/2004/, 2004.

5    McCormack, J., Hoppel, K., Kuhl, D., de Wit, R., Stober, G., Espy, P., Baker, N., Brown, P., Fritts, D., Jacobi, C., Janches, D., Mitchell, N., Ruston, B., Swadley, S., Viner, K., Whitcomb, T., and Hibbins, R.: Comparison of mesospheric winds from a high-altitude meteorological analysis system and meteor radar observations during the boreal winters of 2009–2010 and 2012–2013, Journal of Atmospheric and Solar-Terrestrial Physics, 154, 132–166, https://doi.org/http://dx.doi.org/10.1016/j.jastp.2016.12.007, https://doi.org/10.1016%2Fj.jastp.2016.12.007, 2017.

10    McIntyre, M. E.: On dynamics and transport near the polar mesopause in summer, Journal of Geophysical Research: Atmospheres, 94, 14 617–14 628, https://doi.org/10.1029/JD094iD12p14617, http://dx.doi.org/10.1029/JD094iD12p14617, 1989.

   McLandress, C.: On the importance of gravity waves in the middle atmosphere and their parameterization in general circulation models, Journal of Atmospheric and Solar-Terrestrial Physics, 60, 1357–1383, https://doi.org/10.1016/S1364-6826(98)00061-3, http://dx.doi.org/10.1016/S1364-6826(98)00061-3, 1998.

15    Rapp, M., Strelnikova, I., Latteck, R., Hoffmann, P., Hoppe, U.-P., Häggström, I., and Rietveld, M. T.: Polar mesosphere summer echoes (PMSE) studied at Bragg wavelengths of 2.8m, 67cm, and 16cm, Journal of Atmospheric and Solar-Terrestrial Physics, 70, 947–961, https://doi.org/http://dx.doi.org/10.1016/j.jastp.2007.11.005, http://dx.doi.org/10.1016/j.jastp.2007.11.005, 2008.

   Reid, I. M.: MF and HF radar techniques for investigating the dynamics and structure of the 50 to 110 km height region: a review, Progress in Earth and Planetary Science, 2, https://doi.org/10.1186/s40645-015-0060-7, http://dx.doi.org/10.1186/s40645-015-0060-7, 2015.

20    Singer, W., Latteck, R., Holdsworth, D. A., and Kristiansen, T., eds.: A new narrow beam MF Radar at 3 MHz for studies of the high-latitude middle atmosphere: System description and first results ., 2003.

   Singer, W., Latteck, R., and Holdsworth, D.: A new narrow beam Doppler radar at 3 MHz for studies of the high-latitude middle atmosphere, Advances in Space Research, 41(9), 1488–1494, 2008.

   Sommer, S., Stober, G., and Chau, J. L.: On the angular dependence and scattering model of polar mesospheric summer echoes at VHF,

25    Journal of Geophysical Research: Atmospheres, 121, 278–288, https://doi.org/doi:10.1002/2015JD023518, https://doi.org/10.1002%2F2015jd023518, 2016.

   Stober, G. and Chau, J. L.: A multistatic and multifrequency novel approach for specular meteor radars to improve wind measurements in the MLT region, Radio Science, 50, 431–442, https://doi.org/10.1002/2014RS005591, https://doi.org/10.1002%2F2014rs005591, 2015.

   Stober, G., Latteck, R., Rapp, M., Singer, W., and Zecha, M.: MAARSY – the new MST radar on Andøya: first results of spaced antenna

30    and Doppler measurements of atmospheric winds in the troposphere and mesosphere using a partial array, Advances in Radio Science, 10, 291–298, https://doi.org/10.5194/ars-10-291-2012, http://www.adv-radio-sci.net/10/291/2012/, 2012.

Stober, G., Matthias, V., Jacobi, C., Wilhelm, S., J., H., and Chau, J. L.: Exceptionally strong summer-like zonal wind reversal in the upper mesosphere during winter 2015/16, Annales Geophysicae, 35, 711–720, https://doi.org/10.5194/angeo-35-711-2017, 2017.

35  Suzuki, H., Nakamura, T., Ejiri, M. K., Ogawa, T., Tsutsumi, M., Abo, M., Kawahara, T. D., Tomikawa, Y., Yukimatu, A. S., and Sato, N.: Simultaneous PMC and PMSE observations with a ground-based lidar and SuperDARN HF radar at Syowa Station, Antarctica, Annales Geophysicae, 31, 1793–1803, https://doi.org/doi:10.5194/angeo-31-1793-2013, http://dx.doi.org/10.5194/angeo-31-1793-2013, 2013.

Valentic, T. A., Avery, J. P., Avery, S. K., and Vincent, R. A.: A comparison of winds measured by meteor radar systems and an MF radar at Buckland Park, Radio Science, 32, 867–874, https://doi.org/10.1029/96RS03308, http://dx.doi.org/10.1029/96RS03308, 1997.

Vierinen, J., Chau, J. L., Pfeffer, N., Clahsen, M., and Stober, G.: Coded continuous wave meteor radar, Atmospheric Measurement Techniques, 9, 829–839, https://doi.org/10.5194/amt-9-829-2016, http://dx.doi.org/10.5194/amt-9-829-2016, 2016.

Waldteufel, P. and Corbin, H.: On the Analysis of Single-Doppler Radar Data, Journal of applied meteorology, 18, 532–542, https://doi.org/http://dx.doi.org/10.1175/1520-0450(1979)018<0532:OTAOSD>2.0.CO;2, 1978.

5  Yu, Y., Wan, W., Ren, Z., Xiong, B., Zhang, Y., Hu, L., Ning, B., and Liu, L.: Seasonal variations of MLT tides revealed by a meteor radar chain based on Hough mode decomposition, Journal of Geophysical Research: Space Physics, 120, 7030–7048, https://doi.org/10.1002/2015JA021276, https://doi.org/10.1002%2F2015ja021276, 2015.

| km | 80 | 82 | 84 | 86 | 88 | 90 | 92 | 94 | 96 | 98 |
|---|---|---|---|---|---|---|---|---|---|---|
| Andenes | 0.57 | 0.56 | 0.52 | 0.42 | 0.21 | -0.13 | -0.45 | -0.61 | -0.67 | -0.69 |
| Juliusruh | 0.43 | 0.36 | 0.23 | 0.04 | -0.23 | -0.48 | -0.62 | -0.67 | -0.68 | -0.68 |
| Collm | 0.3 | 0.19 | -0.01 | -0.3 | -0.54 | -0.65 | -0.68 | -0.68 | -0.66 | -0.64 |
| Davis | -0.37 | -0.37 | -0.38 | -0.39 | -0.41 | -0.42 | -0.41 | -0.38 | -0.35 | -0.32 |

**Table 1.** Correlation coefficients between the zonal wind and the LOD. Positive values corresponds to the occurrence of e.g., an eastward directed  mean zonal wind together with a positive fluctuation in the LOD.

[Figure]

Equinox

Aphelion
July 4

Perihelion
January 3

152.100.000 km    SUN    147.100.000 km

E

E

E

Equinox

E

- farther Earth-Sun distance
- shrinking of the atmosphere

- faster Earth rotation
- decreased LOD

- shorter Earth-Sun distance
- expansion of the atmosphere

- slower Earth rotation
- increased LOD

**Figure 1.** Schema of Earth and Sun correlation and the resulting effects on the thickness of the atmosphere and the Earth's rotation velocity.

[Figure]

**Figure 2.** Composites of zonal wind for the northern hemisphere stations Andenes (top), Juliusruh (2nd row), and Collm (3th row). At the bottom is shown the southern hemispheric station of Davis. The composite for Andenes, Collm, and Davis include 12 years of meteor radar data and that of Juliusruh 9 years. Positive values correspond to eastward directed winds and negative to westward directed winds.

[Figure]

**Figure 3.** Composite of zonal wind for high latitude location (top), and mid latitude location (bottom). The composite of both figures includes 12 years of data wind data derived from MLS geopotential height data. Positive values corresponds to eastward directed winds and negative to westward directed winds. The altitude is given in geopotential height.

[Figure]

**Figure 4.** Theoretical change of the rotation speed (left side) for a rigid atmospheric layer. In black the theoretical rotation speed of the Earth's atmosphere and in colors the change due to density increase of 1% according the legend. On the right side the density progress is shown for specific altitudes.

[Figure]

**Figure 5.** Zonal wind amplitudes for winter and summer season at 96 km and 88 km for Andenes and Davis.

[Figure]

**Figure 6.** Smoothed zonal wind (blue) values based on meteor radar wind data at 80 km and smoothed LOD (black) values. The modulation of the smoothed zonal wind is displayed in red after removing the impact of the solar cycle, whereby the smoothing is stronger as in blue. All curves are done by a smooth over several days, without removing the day-to-day variations, to show the seasonal pattern of the parameters. The dashed lines corresponds to the tendency of the wind/LOD based on linear regression.

[Figure]

**Figure 7.** Same as Figure **??**, but for 96 km.

[Figure]

**Figure 8.** Zonal MLS wind (red) and LOD (black) at ∼80 km geometric height for 0°-20°E. The left part show the values for the southern hemisphere, the right for the northern hemisphere, for every 10° latitude. The black correlation coefficients (r) are estimated for the mean between 0°E and 20°E, and the green coefficients corresponds to global average over all longitudes.

[Figure]

**Figure 9.** Annual mean values for the LOD (black) and the zonal wind (red), for the station Collm, after removing seasonal variations and the solar cycle for the altitudes between 80 and 100 km. The errorbars corresponds to the standard deviation. The dashed lines represents the tendency.

[Figure]

**Figure 10.** Same as Figure **??**, but for Davis.